# Impacts of biogenic polyunsaturated aldehydes on metabolism and community composition of particle-attached bacteria in coastal hypoxia

Zhengchao Wu[1,2], Qian P. Li[1,2,3,]*, Zaiming Ge[1,3], Bangqin Huang[4], Chunming Dong[5]

[1]State Key Laboratory of Tropical Oceanography, South China Sea Institute of Oceanology, Chinese Academy of Sciences, Guangzhou, China

[2]Southern Marine Science and Engineering Guangdong Laboratory, Guangzhou, China

[3]College of Marine Science, University of the Chinese Academy of Sciences, Beijing, China

[4]Fujian Provincial Key Laboratory of Coastal Ecology and Environmental Studies, State Key Laboratory of Marine Environmental Science, Xiamen University, Xiamen, China

[5]Key Laboratory of Marine Genetic Resources, Third Institute of Oceanography, MNR, Xiamen, China

*Correspondence to*: Qian Li (qianli@scsio.ac.cn)

**Abstract.** Eutrophication-driven coastal hypoxia is of great interest for decades, though its mechanisms remain not fully understood. Here, we showed elevated concentrations of particulate and dissolved polyunsaturated aldehydes (PUAs) associated with the hypoxic waters in the bottom layer of a salt-wedge estuary. Bacterial respiration within the hypoxic waters was mainly contributed by particle-attached bacteria (PAB) ($>0.8$ μm), with free-living bacteria (0.2-0.8 μm) only accounting for 25-30 % of the total rate. The concentration of particle-adsorbed PUAs ($\sim$10 μmol $L^{-1}$) in the hypoxic waters were directly quantified for the first time based on large-volume-filtration and subsequent on-site PUAs derivation and extraction. PUAs-amended incubation experiments for PAB ($>25$ μm) associated with sinking or suspended particles retrieved from the low-oxygen waters were also performed to explore the impacts of PUAs on the growth and metabolism of PAB and associated oxygen utilization. We found an increase in cell growth of PAB in response to low-dose PUAs (1 μmol $L^{-1}$) but an enhanced cell-specific bacterial respiration and production in response to high-dose PUAs (100 μmol $L^{-1}$). Improved cell-specific metabolism of PAB in

response to high-dose PUAs was also accompanied by a shift of PAB community structure with increased
dominance of genus *Alteromonas* within the Gammaproteobacteria. We thus conclude that a high PUAs
concentration associated with aggregate particles within the bottom layer may be crucial for some species
within *Alteromonas* to regulate PAB community structure. The change of bacteria community could lead to
an enhancement of oxygen utilization during the degradation of particulate organic matters and thus likely
contribute to the formation of coastal hypoxia. These findings are potentially important for coastal systems
with large river inputs, intense phytoplankton blooms driven by eutrophication, as well as strong hypoxia
developed below the salt-wedge front.

## 1. Introduction

Coastal hypoxia, defined as dissolved oxygen levels < 62.5 μmol kg$^{-1}$, has become a worldwide problem in recent decades (Diaz and Rosenberg, 2008; Helm et al., 2011). It could affect diverse life processes from genes to ecosystems, resulting in the spatial and temporal change of marine food-web structures (Breitburg et al., 2018). Coastal deoxygenation is also tightly coupled with other global issues, such as global warming and ocean acidification (Doney et al., 2012). Formation and maintenance of eutrophication-derived hypoxia in the coastal waters should reflect the interaction between physical and biogeochemical processes (Kemp et al., 2009). Generally, seasonal hypoxia occurs in the coastal ocean when strong oxygen sinks are coupled with restricted resupply during periods of strong density stratification. Termination of the event occurs with oxygen resupply when stratification is eroded by vertical mixing (Fennel and Testa, 2019).

Bacterial respiration accounts for the largest portion of aquatic oxygen consumption and is thus pivotal for the development of hypoxia and oxygen minimum zones (Williams and del Giorgio, 2005; Diaz and Rosenberg, 2008). Generally, free-living bacteria (FLB, 0.2-0.8 μm) dominate the community respiration in many parts of the ocean (Robinson and Williams, 2005; Kirchman, 2008). Compared to the FLB, the role of particle-attached bacteria (PAB, >0.8 μm) on community respiration is less addressed, particularly in the coastal oceans. In some coastal waters, PAB can be more important than the FLB with a higher metabolic activity that might affect carbon cycle through organic matter remineralization (Garneau et al., 2009; Lee et al., 2015). PAB was found more abundant than the FLB with a higher diversity near the mouth of the Pearl River estuary (PRE) (Li et al., 2018; Liu et al., 2020; Zhang et al., 2016). An increased contribution of PAB to respiration relative to FLB can occur during the development of coastal phytoplankton bloom (Huang et al., 2018). In the Columbia River estuary, the particle-attached bacterial activity could be 10-100 folds higher than that of its free-living counterparts leading to its dominant role in organic detritus remineralization (Crump et al., 1998). Therefore, it is crucial to assess the respiration process associated with PAB and its controlling factors in these regions, to fully understand oxygen utilization in the hypoxic area with an intense supply of particulate organic matters.

There is an increasing area of seasonal hypoxia in the nearshore bottom waters of the Pearl River
Estuary and the adjacent northern South China Sea (NSCS) (Yin et al., 2004; Zhang and Li 2010; Su et al.,
2017). The hypoxia is generally developed at the bottom of the salt-wedge where downward mixing of
oxygen is restrained due to increased stratification and where there is an accumulation of
eutrophication-derived organic matter due to flow convergence driven by local hydrodynamics (Lu et al.,
2018). Besides physical and biogeochemical conditions, aerobic respiration is believed the ultimate cause
of hypoxia here (Su et al., 2017). Thus, microbial respiration had been strongly related to the consumption
of bulk dissolved organic carbon in the PRE hypoxia (He et al., 2014).
Phytoplankton-derived polyunsaturated aldehydes (PUAs) are known to affect marine microorganisms
over various trophic levels by acting as infochemicals and/or by chemical defenses (Ribalet et al., 2008;
Ianora and Miralto, 2010; Edwards et al., 2015; Franzè et al., 2018). PUAs are produced by stressed
diatoms during the oxidation of membrane polyunsaturated fatty acids (PUFA) by lipoxygenase (Pohnert
2000) and are released from the surface of particles to the seawater by diffusion. A perennial bloom of
PUA-producing diatoms near the mouth of the PRE (Wu and Li, 2016) should support the importance of
PUAs relative to other phytoplankton-derived organic compounds, such as karlotoxin by dinoflagellates,
cyanotoxin by cyanobacteria, and dimethylsulphoniopropionate mainly by prymnesiophytes. Besides PUAs,
there are other signaling molecules that may potentially affect bacterial activities in the low oxygen waters,
such as 2-n-pentyl-4-quinolinol (PQ) and acylated homoserine lactones (AHL). PQ could be less important
here in terms of hypoxia formation as it is generally produced as antibiosis by PAB such as *Alteromonas sp.*
to inhibit respirations of other PAB (Long et al., 2003). AHL could also play a less important role here since
the AHL-mediated quorum-sensing could be constrained by a large pH fluctuation from 7.2 to 8.8 in the
bottom waters of the PRE (Decho et al., 2009).
The level of PUAs in the water-column are inhomogeneous, varying from sub-nanomolar offshore to
nanomolar nearshore (Vidoudez et al., 2011; Wu and Li, 2016; Bartual et al., 2018), and to micromolar
associated with particle hotspots (Edwards et al., 2015). The strong effect of PUAs on bacterial growth,
production, and respiration has been well demonstrated in laboratory studies (Ribalet et al., 2008) and field
studies (Balestra et al., 2011; Edwards et al., 2015). A nanomolar level of PUAs recently reported in the
coastal waters outside the PRE was hypothesized to affect oxygen depletion by promoting microbial
utilization of organic matters in the bottom waters (Wu and Li, 2016). Meanwhile, the actual role of PUAs
on bacterial metabolism within the bottom hypoxia remains largely unexplored.
In this study, we investigate the particle-attached bacteria within the core of the hypoxic waters by
exploring the linkage between PUAs and bacterial oxygen utilization on the suspended organic particles.
There are three specific questions to address here: What are the relative roles of PAB and FLB on bacterial
respiration in the hypoxic waters? What are the actual levels of PUAs in the hypoxic waters? What are the
responses of PAB to PUAs in the hypoxic waters? For the first question, size-fractionated bacterial
respiration rates were estimated for both FLB (0.2-0.8 μm) and PAB (>0.8 μm) in the hypoxic waters. For
the second question, the concentrations of particulate and dissolved PUAs within the hypoxic waters were
measured in the field. Besides, the hotspot PUAs concentration associated with the suspended particles
within the hypoxic waters was directly quantified for the first time using large-volume filtration and
subsequent on-site derivation and extraction. For the third question, field PUAs-amended incubation
experiments were conducted for PAB (>25 μm) retrieved from the low-oxygen waters. We focused on
particles of >25 μm to better explore the role of PUAs on PAB associated with sinking or suspended
particles. The doses of PUAs treatments were selected to represent the actual levels of PUAs hotspots, to
assess the PAB responses (including bacterial abundance, respiration, production, and community
composition) to the exogenous PUAs in the hypoxic waters. By synthesizing these experimental results
with the change of water-column biogeochemistry, we hope to explore the underlying mechanism for
particle-adsorbed PUAs influencing on community structure and metabolism of PAB in the low-oxygen
waters, as well as to understand its contribution to coastal deoxygenation of the NSCS shelf-sea.

**2. Methods**

## 2.1 Descriptions of field campaigns and sampling approaches

Field survey cruises were conducted in the PRE and the adjacent NSCS during June 17th-28th, 2016 and

June 18st-July 2nd, 2019 (Figure 1). Briefly, vertical profiles of temperature, salinity, dissolved oxygen, and

turbidity were acquired from a Seabird 911 rosette sampling system. The oxygen sensor data were corrected

by field titration measurements during the cruise. Water samples at various depths were collected using 6 or

12 liters (12 or 24 positions) Niskin bottles attached to the Rosette sampler. Surface water samples were

collected at ~1m or 5 m depth, while bottom water samples were obtained at depths ~4 m above the bottom.

Chlorophyll-*a* (Chl-*a*) samples were taken at all depths at all stations and nutrients were also sampled

except at a few discrete stations. For the 2016 cruise, samples for pPUAs were collected at all depths close

to station X1 (Figure 1A). During the summer of 2019, vertical profiles of particulate PUAs (pPUAs) and

dissolved PUAs (dPUAs) were determined at Y1 in the hypoxic zone and Y2 outside the hypoxic zone with

field PUAs-amended experiments conducted at Y1 (Figure 1B). For station Y1, the middle layer was

defined as 12 m with the bottom layer as 25 m. At this station, samples at different depths were collected

for determining the size-fractionated respiration rates and the whole water bacterial taxonomy.

## 2.2 Determination of chlorophyll-*a*, dissolved nutrients

For Chl-*a* analyses, 500 mL of water sample was gently filtered through a 0.7 μm Whatman GF/F filter.

The filter was then wrapped by a piece of aluminum foil and stored at -20 $^\circ$C on board. Chl-*a* was extracted

at 4 ºC in the dark for 24 h using 5 mL of 90% acetone. After centrifuged at 4000 rpm for 10 min, Chl-*a*

was measured using a standard fluorometric method with a Turner Designs fluorometer (Parsons et al.,

1984). Water samples for nutrients were filtered through 0.45 μm Nucleopore filters and stored at -20 $^\circ$C.

Nutrient concentrations including nitrate plus nitrite, phosphate, and silicate were measured using a

segmented-flow nutrient autoanalyzer (Seal AA3, Bran-Luebbe, GmbH).

## 2.3 Sampling and measurements of particulate and dissolved PUAs in one-liter seawater

We used a similar protocol of Wu and Li (2016) for pPUAs and dPUAs collection, pretreatment, and

determination. Briefly, 2-4 liters of water sample went through a GF/C filtration with both the filter and the

filtrate collected separately. The filter was rinsed by the derivative solution with the suspended particle

samples collected in a glass vial. After adding internal standard, the samples in the vial were frozen and

thawed three times to mechanically break the cells for pPUAs. The filtrate from the GF/C filtration was

also added with internal standard and transferred to a C18 solid-phase extraction cartridge. The elute from

the cartridge with the derivative solution was saved in a glass vial for dPUAs. Both pPUAs and dPUAs

samples were frozen and stored at -20 ºC.

In the laboratory, the pPUAs sample was thawed with the organic phase extracted. After the solvent

was evaporated with the sample concentrated and re-dissolved in hexane, pPUAs was determined using gas

chromatography and mass spectrometry (Agilent Technologies Inc., USA). Standards series were prepared

by adding certain amounts of three major PUAs to the derivative solution and went through the same

pretreatment and extraction steps as samples. Derivatives of dPUAs were extracted and measured by

similar methods as pPUAs, except that the calibration curves of dPUAs were constructed separately. The

units of pPUAs and dPUAs are nmol $L^{-1}$ (nmol PUA in one-liter seawater).

**2.4 Particle collections by large-volume filtrations in hypoxia waters.**

Large volumes (~300 L) of the middle (12 m) and the bottom (25 m) waters within the hypoxia zone were

collected by Niskin bottles at station Y1. For each layer, the water sample was quickly filtered through a

sterile fabric screen (25 μm filter) on a disk filter equipped with a peristaltic pump to qualitatively obtain

particles of >25 μm. Larger zooplankters were picked off immediately. The particle samples were gently

back-flushed three times off the fabric screen using particle-free seawater (obtained using a 0.2 μm

filtration of the same local seawater) into a sterile 50-mL sampling tube.

The volume of total particles from large-volume-filtration was measured as follows: The collected

particle in the 50 mL tube was centrifuged for one minute at a speed of 3000 revolutions per minute (r.p.m)

with the supernatant saved (Hmelo et al., 2011). The particle sample was resuspended as slurry by gently
shaking and transferred into a sterile 5 mL graduated centrifuge tube. The sample was centrifuged again by
the same centrifuging speed with the final volume of the total particles recorded. The unit for the total
particle volume is mL.
All the particles were transferred back to the sterile 50 mL centrifuge tube (so as all the supernatants)
with 0.2-µm-filtered seawater, which was used for subsequent measurements of particle-adsorbed PUAs as
well as for PUAs-amended incubation experiments of particle-attached bacteria.

**2.5 Measurements of particle-adsorbed PUAs**
After gently shaking, 3 mL of sample in the 50 mL sampling tube (see section 2.4) was used for the
analyses of particle-adsorbed PUAs concentration (two replicates) according to the procedure shown in
Figure 2 (modified from the protocols of Edwards et al. 2015 and Wu and Li 2016). The sample (3 mL) was
transferred to 50 mL centrifuge tubes for PUAs derivatization on board. An internal standard of
benzaldehyde was added to obtain a final concentration of 10 µM. The aldehydes in the samples were
derivatized by the addition of O-(2,3,4,5,6-pentafluorobenzyl) hydroxylamine hydrochloride solution in
deionized water ($p$H=7.5). The reaction was performed at room temperature for 15 min (shaking slightly
for mix every 5 min). Then 2 mL sulfuric acid (0.1%) solution was added to a final concentration of 0.01%
acid ($p$H of 2-3) to avoid new PUAs induced by enzymatic cascade reactions. The derivate samples were
subsequently sonicated for 3 min before the addition of 20 mL hexane, and the upper organic phase of the
extraction was transferred to a clean tube and stored at -20 °C.
Upon returning to the laboratory, the adsorbed PUAs on these particles (undisrupted PUAs) were
determined with the same analytical methods as those for the disrupted pPUAs (freeze-thaw methods to
include the portion of PUAs eventually produced as cells die, Wu and Li 2016) except for the freeze-thaw
step. A separate calibration curve was made for the undisrupted PUAs derivates. A standard series of
heptadienal, octadienal, and decadienal (0, 0.1, 0.5, 1.0, 2.5, 5.0, 10.0, 25.0 nmol $L^{-1}$) was prepared before

each analysis by diluting a relevant amount of the PUA stock solution (methanolic solution) with deionized water. These standard solutions were processed through all the same experimental steps as those mentioned above for derivation, extraction, and measurement of the undisrupted PUAs sample. The unit for the undisrupted PUAs is nmol $L^{-1}$. The total amount of the undisrupted PUAs in the 50 mL sampling tube was the product of the measured concentration and the total volume of the sample.

The hotspot PUAs concentration associated with the aggregate particles is defined as the PUAs concentration in the volume of the water parcel displaced by these particles. Therefore, the final concentration of particle-adsorbed PUAs in the water column, defined as PUAs [$\mu$mol $L^{-1}$], should be equal to the moles of particle-adsorbed PUAs (nmol, the undisrupted PUAs) divided by the volume of particles (mL).

## 2.6 Incubation of particle-attached bacteria with PUAs treatments.

The impact of PUAs on microbial growth and metabolisms in the hypoxia zone was assessed by field incubation of particle-attached bacteria on particles of > 25 $\mu$m collected from large-volume filtration with direct additions of low or high doses of PUAs (1 or 100 $\mu$mol $L^{-1}$) on June, 29[th], 2019 (Figure 2).

A sample volume of ~32 mL in the centrifuge tube (section 2.4) was transferred to a sterile Nalgene bottle before being diluted by particle-free seawater to a final volume of 4 L. About 3.2 L of the sample solution was transferred into four sterile 1-L Nalgene bottles (each with 800 mL). One 1-L bottle was used for determining the initial conditions: after gentle shaking, the solution was transferred into six biological oxygen demand (BOD) bottles with three for initial oxygen concentration (fixed immediately by Winkler reagents) and the other three for initial bacterial abundance, production, and community structure. The other three 1-L bottles were used for three different treatments (each with two replicates in two 0.5-L bottles): the first one served as the control with the addition of 200 $\mu$L methanol, the second one with 200 $\mu$L low-dose PUAs solution, and the third one with 200 $\mu$L high-dose PUAs solution (Table 1). The solution in each of the three treatments (0.5L bottles) was transferred to six parallel replicates by 60-mL

BOD bottles. These BOD bottles were incubated at *in situ* temperature in the dark for 12 hours. At the end
of each incubation experiment, three of the six BOD bottles were used for determining the final oxygen
concentrations with the other three for the final bacterial abundance, production, and community structure.

211         To test the possibility of PUAs as carbon sources for bacterial utilization, a minimal medium was

prepared with only sterile artificial seawater but not any organic carbons (Dyksterhouse et al., 1995). A
volume of 375 μL sample (from the above 4 L sample solution) was inoculated in the minimal medium
amended with heptadienal in a final concentration of about 200 μmol $L^{-1}$. This PUA level was close to the
hotspot PUAs of 240 μmol $L^{-1}$ found in the suspended particles of a station near the PRE. It was also
comparable to the hotspot PUAs of 25.7 μmol $L^{-1}$ in the temperate west North Atlantic (Edwards et al.,
2015). For comparisons, the same amount of sample was also inoculated in the minimal medium (75 mL)
amended with an alkane mixture (ALK, n-pentadecane and n-heptadecane) at a final concentration of 0.25
g $L^{-1}$, or with a mixture of polycyclic aromatic hydrocarbons (PAH, naphthalene and phenanthrene) at a
final concentration of 200 ppm. These experiments were performed in dark at room temperature for over 30
days. Significant turbidity changes in the cell culture bottle over incubation time will be observed if there is
a carbon source for bacterial growth.

**2.7 Measurements of bacteria-related parameters**
**(1) Bacterial abundance**
At the end of the 12-h incubation period, a 2 mL sample from each BOD bottle was preserved in 0.5%
glutaraldehyde. The fixation lasted for half of an hour at room temperature before being frozen in liquid $N_2$
and stored in a −80 °C freezer. In the laboratory, the samples were performed through a previously
published procedure for detaching particle-attached bacteria (Lunau et al., 2005), which had been proved
effective for samples with high particle concentrations. To break up particles and attached bacteria, 0.2 mL
pure methanol was added to the 2 mL sample and vortexed. The sample was then incubated in an ultrasonic
bath (35 kHz, 2 x 320W per period) at 35 °C for 15 min. Subsequently, the tube sample was filtered with a
50 µm-filter to remove large detrital particles. The filtrate samples for surface-associated bacteria cells
were diluted by 5-10 folds using TE buffer solution and stained with 0.01% SYBR Green I in the dark at
room temperature for 40 min. With the addition of 1-µm beads, bacterial abundance (BA) of the samples
was counted by a flow cytometer (Beckman Coulter CytoFlex S) with bacteria detected on a plot of green
fluorescence versus side scatter (Marie et al., 1997). The precision of the method estimated by the
coefficient of variation (CV%) was generally less than 5%.
For bulk-water bacteria abundance, 1.8 mL of seawater sample was collected after a 20-µm
prefiltration. The sample was transferred to a 2 mL centrifuge tube and fixed by adding 20 µL of 20%
paraformaldehyde before storage in a −80 °C freezer. In the laboratory, 300 µL of the sample after thawing
was used for staining with SYBR Green and analyzed using the same flow cytometry method as above
(Marie, et al, 1997).

**(2) Bacterial respiration**
For BOD samples, bacterial respiration (BR) was calculated based on the oxygen decline during the 12-h
incubation and was converted to carbon units with the respiratory quotient assumed equal to 1 (Hopkinson,
1985). Dissolved oxygen was determined by a high-precision Winkler titration apparatus (Metrohm-848,
Switzerland) based on the classic method (Oudot et al., 1988). We should mention that BR could be
overestimated if phytoplankton and microzooplankton were present in the particle aggregates of $> 25$ µm.
However, this effect could be relatively small because the raw seawater in the hypoxic zone had very low
chlorophyll-*a* and because there was virtually not much microzooplankton in the sample (confirmed by
FlowCAM).
Method for the estimation of the bulk water bacterial respiration at stations X1, X2, and X3 can be
found in Xu et al (2018). For the bulk water at station Y1, the size-fractionated respiration rates, including
free-living bacteria of 0.2-0.8 µm and particle-associated community of >0.8 µm (we assumed that they
were mostly PAB given the low phytoplankton chlorophyll-*a* of the sample and the absence of zooplankton
during the filtration), were estimated based on the method of García-Martín et al (2019). Four 100 mL
polypropylene bottles were filled with seawater. One bottle was immediately fixed by formaldehyde. After
15 min, the sample in each bottle was incubated in the dark at the *in situ* temperature after the addition of
the Iodo-Nitro-Tetrazolium (INT) salt at a final concentration of 0.8 mmol L$^{-1}$. The incubation reaction
lasted for 1.5 h before being stopped by formaldehyde. After 15 min, all the samples were sequentially
filtered through 0.8 and 0.2 μm pore size polycarbonate filters and stored frozen until further measurements
by spectrophotometry.

**(3) Bacterial production**
Bacterial production (BP) was determined using a modified protocol of the $^{3}$H-leucine incorporation
method (Kirchman, 1993). Four 1.8-mL aliquots of the sample were collected by pipet from each BOD
incubation and added to 2-mL sterile microcentrifuge tubes, which were incubated with $^{3}$H-leucine (in a
final concentration of 4.65 μmol Leu L$^{-1}$, Perkin Elmer, USA). One tube served as the control was fixed by
adding 100% trichloroacetic acid (TCA) immediately (in a final concentration of 5%). The other three were
terminated with TCA at the end of the 2-h dark incubation. Samples were filtered onto 0.2-μm
polycarbonate filters and then rinsed twice with 5% TCA and three times with 80% ethanol (Huang et al.,
2018) before being stored at −80 °C. In the laboratory, the filters were transferred to scintillation vials with
5 mL of Ultima Gold scintillation cocktail. The incorporated $^{3}$H was determined using a Tri-Carb 2800TR
liquid scintillation counter. Bacterial production was calculated with the previous published
leucine-to-carbon empirical conversion factors of 0.37 kg C mol leucine$^{-1}$ in the study area (Wang et al.,
2014). Bacterial carbon demand (BCD) was calculated as the sum of BP and BR. Bacterial growth
efficiency (BGE) was equated to BP/BCD.

**(4) Bacterial community structure**

At the end of incubation, the DNA sample was obtained by filtering 30 mL of each BOD water via a 0.22-μm Millipore filter, which was preserved in a cryovial with the DNA protector buffer and stored at -80 °C. DNA was extracted using the DNeasy PowerWater Kit with genomic amplification by Polymerase Chain Reaction (PCR). The V3 and V4 fragments of bacterial 16S rRNA were amplified at 94 °C for 2 min and followed by 27 cycles of amplification (94 °C for 30 s, 55° C for 30 s, and 72 °C for 60 s) before a final step of 72 °C for 10 min. Primers for amplification included 341F (CCTACGGGNGGCWGCAG) and 805R (GACTACHVGGGTATCTAATCC). Reactions were performed in a 10-μL mixture containing 1 μL Toptaq Buffer, 0.8 μL dNTPs, 10 μM primers, 0.2 μL Taq DNA polymerase, and 1 μL Template DNA. Three parallel amplification products for each sample were purified by an equal volume of AMpure XP magnetic beads. Sample libraries were pooled in equimolar and paired-end sequenced (2×250 bp) on an Illumina MiSeq platform.

High-quality sequencing data was obtained by filtering on the original off-line data. Briefly, the raw data was pre-processed using TrimGalore to remove reads with qualities of less than 20 and FLASH2 to merge paired-end reads. Besides, the data were also processed using Usearch to remove reads with a total base error rate of greater than 2 and short reads with a length of less than 100 bp and using Mothur to remove reads containing more than 6 bp of N bases. We further used UPARSE to remove the singleton sequence to reduce the redundant calculation during the data processing. Sequences with similarity greater than 97% were clustered into the same operational taxonomic units (OTUs). R software was used for community composition analysis.

DNA samples for the bulk bacteria (>0.2 μm) and PAB on particles of > 25 μm at station Y1 were also collected for bacterial community analysis using the same method described above. Methods for the bulk water bacterial community analyses at stations X1, X2, and X3 during the 2016 cruise can be found in the published paper of Xu et al. (2018).

**2.8 Statistical Analysis**

All statistical analyses were performed using the statistical software SPSS (Version 13.0, SPSS Inc.,

Chicago, IL, USA). A student's t-test with a 2-tailed hypothesis was used when comparing PUAs-amended

treatments with the control or comparing stations inside and outside the hypoxic zone, with the null

hypothesis being rejected if the probability ($p$) is less than 0.05. We consider $p$ of <0.05 as significant and $p$

of <0.01 as strong significant. Ocean Data View with the extrapolation model "DIVA Gridding" method

was used to contour the spatial distributions of physical and biogeochemical parameters.

## 3. Results

### 3.1 Characteristics of hydrography, biogeochemistry, and bulk bacteria community in the hypoxic zone

During our study periods, there was a large body of low oxygen bottom water with the strongest hypoxia (<

62.5 μmol kg$^{-1}$) on the western shelf of the PRE (Figure 1), which was relatively similar among different

summers of 2016 and 2019 (Figure 1). For vertical distribution, a strong salt-wedge structure was found

over the inner shelf (Figures 3A, 3D) with freshwater on the shore side due to intense river discharge.

Bottom waters with oxygen deficiency (< 93.5 μmol kg$^{-1}$) occurred below the lower boundary of the

salt-wedge and expanded ~60 km offshore (Figure 3E). In contrast, a surface high Chl-$a$ patch (6.3 μg L$^{-1}$)

showed up near the upper boundary of the front, where there was enhanced water-column stability, low

turbidity, and high nutrients (Figures 3B, 3C). Therefore, there was a spatial mismatch between the

subsurface hypoxic zone (Figure 3E) and the surface chlorophyll-bloom (Figure 3F) during the

estuary-to-shelf transect, as both the surface Chl-$a$ and oxygen right above the hypoxic zones at the bottom

boundary of the salt-wedge were not themselves maxima.

There were much higher rates of respiration (BR) ($t$=7.8, $n$=9, $p$<0.01) and production (BP) ($t$=13.0,

$n$=9, $p$<0.01) for the bulk bacterial community (including FLB and PAB) in the bottom waters of X1 within

the hypoxic core than those of X2 and X3 outside the hypoxic zone during June 2016 (Figure 4, modified

from data of Xu et al., 2018). The size-fractionated respiration rates were quantified at station Y1 during the

2019 cruise (Figure S1) to distinguish the different roles of FLB and PAB on bacterial respiration in the
hypoxic waters. Our results suggested that bacterial respiration within the hypoxic waters was largely
contributed by PAB (>0.8 μm), which was about 2.3-3 folds of that by FLB (0.2-0.8 μm).
The bulk bacterial composition of the bottom water of X1 during the 2016 cruise with 78% of
α-Proteobacteria (α-Pro), 15% of γ-Proteobacteria (γ-Pro), and 6% of Bacteroidetes was significantly
different from those of X2 and X3 (91% α-Pro, 5% γ-Pro, and 2% Bacteroidetes), although their bacterial
abundances were about the same (Figure 4). Compared to that of the 2016 cruise, there was a different
taxonomic composition of the bulk bacterial community in the hypoxic waters of the 2019 cruise with on
average 33% of α-Pro, 25% of γ-Pro, and 14% of Bacteroidetes. Furthermore, there was a substantially
different taxonomic composition for PAB (>25 μm) with on average 66% of γ-Pro, 22% of α-Pro, and 4%
of Bacteroidetes (Figure S2A). In particular, there was an increase of γ-Pro, but a decrease of α-Pro and
Bacteroidetes, in the PAB (>25 μm) relative to the bulk bacterial community. On the genus level, the PAB
(>25 μm) was largely dominated by the *Alteromonas* group in both the middle and bottom waters (Figure
S2B).

**3.2 PUAs concentrations in the hypoxic zone**

Generally, there were significantly higher pPUAs of 0.18 nmolL$^{-1}$ ($t$=3.20, $n$=10, $p$<0.01) and dPUAs of
0.12 nmol L$^{-1}$ ($t$=7.61, $n$=8, $p$<0.01) in the hypoxic waters than in the nearby bottom waters without
hypoxia (0.02 nmol L$^{-1}$ and 0.01 nmol L$^{-1}$). Vertical distributions of pPUAs and dPUAs in the bulk seawater
were showed for two stations (Y1 and Y2) inside and outside the hypoxic zone (Figure 1). Nanomolar
levels of pPUAs and dPUAs were found in the water column in both stations (Figures 5E, 5F). There were
high pPUAs and dPUAs in the bottom hypoxic waters of station Y1 (Figure 5E, 5F) together with locally
elevated turbidity (Figure 3B) when compared to the bottom waters outside, which likely a result of particle
resuspension. For station Y2 outside the hypoxia, we found negligible pPUAs and dPUAs at depths below
the mixed layer (Figure 5E, 5F), which could be due to PUAs dilution by the intruded subsurface seawater.
Particle-adsorbed PUAs in the low-oxygen waters were quantified for the first time with the direct
particle volume estimated by large-volume-filtration (see the method section), which would reduce the
uncertainty associated with particle volume calculated by empirical equations derived for marine-snow
particles (Edward et al., 2015). We found high levels of particle-adsorbed PUAs ($\sim$10 $\mu$mol L$^{-1}$) in these
waters (Figure 6), which were orders of magnitude higher than the bulk water pPUAs or dPUAs
concentrations (<0.3 nmol L$^{-1}$, Figure 5E, 5F). Particle-adsorbed PUAs of the low-oxygen waters mainly
consisted of heptadienal (C7_PUA) and octadienal (C8_PUA).

**3.3 Particle-attached bacterial growth and metabolism in the hypoxic zone**
Incubation of the PAB acquired from the low-oxygen waters with direct additions of different doses of
exogenous PUAs over 12 hours was carried out to examine the change of bacterial growth and metabolism
activities in response to PUA-enrichments. At the end of the incubation experiments, BA was not different
from the control for the PH treatment (Figure 7A). However, for the PL treatment, there were substantial
increases of BA in both the middle and the bottom waters compared to the initial conditions (Figure 7A). In
particular, BA of $\sim$3.2 $\pm$ 0.04 $\times$ 10$^9$ cells L$^{-1}$ in the bottom water for the PL treatment was significantly
higher ($t$=12.26, $n$=12, $p$<0.01) than the control of 2.5 $\pm$ 0.07 $\times$ 10$^9$ cells L$^{-1}$.
BR was significantly promoted by the low-dose PUAs with a 21.6% increase in the middle layer
($t$=11.91, $n$=8, $p$<0.01) and a 25.8% increase in the bottom layer ($t$=11.50, $n$=8, $p$<0.01) compared to the
controls. Stimulating effect of high-dose PUAs on BR was even stronger with 47.0% increase in the middle
layer ($t$=30.56, $n$=8, $p$<0.01) and 39.8% increase in the bottom layer ($t$=9.40, $n$=8, $p$<0.01) (Figure 7B).
Meanwhile, the cell-specific BR was significantly improved for both layers with high-dose of PUAs
($t$=15.13, $n$=8, $p$<0.01 and $t$=4.77, $n$=8, $p$<0.01), but not with low-dose of PUAs (Figure 7C) due to
increase of BA (Figure 7A). BGE was generally very low (<1.5%) during all the experiments (Figure 7D)
due to substantially high rates of BR (Figure 7B) than BP (Figure 7E). Also, there was no significant
difference in BGE between controls and PUAs treatments for both layers (Figure 7D).
For the bottom layer, BP was $12.6 \pm 0.8$ µg C $L^{-1}$ $d^{-1}$ for low-dose PUAs and $16.4 \pm 0.6$ µg C $L^{-1}$ $d^{-1}$
for high-dose PUAs, which were both significantly ($t=2.98$, $n=8$, $p<0.05$ and $t=10.41$, $n=8$, $p<0.01$) higher
than the control of $10.6 \pm 0.6$ µg C $L^{-1}$ $d^{-1}$. Meanwhile, BP in the middle layer was significantly higher
($t=2.52$, $n=8$, $p<0.05$) than the control for high-dose PUAs ($13.4 \pm 0.9$ µg C $L^{-1}$ $d^{-1}$) but not for low-dose
PUAs ($12.6 \pm 0.9$ µg C $L^{-1}$ $d^{-1}$) (Figure 7E). The cell-specific BP (sBP, $7.9 \pm 0.5$ and $6.9 \pm 0.2$ fg C $cell^{-1}$ $d^{-1}$)
for high-dose PUAs were significantly ($t=2.62$, $n=8$, $p<0.05$ and $t=11.26$, $n=8$, $p<0.01$) higher than the
control in both layers (Figure 7F). Meanwhile, for low-dose PUAs, the sBP in both layers were not
significantly different from the controls.

**3.4 Particle-attached bacterial community change during incubations**
Generally, γ-Pro dominated (>68%) the bacterial community at the class level for all experiments, followed
by the second largest bacterial group of α-Pro. There was a significant increase of γ-Pro by high-dose PUAs
with increments of 17.2% ($t=9.25$, $n=8$, $p<0.01$) and 19.5% ($t=6.32$, $n=8$, $p<0.01$) for the middle and the
bottom layers, respectively (Figure 8A). However, there was no substantial change of bacterial community
composition by low-dose PUAs for both layers (Figure 8A).
On the genus level, there was also a large difference in the responses of various bacterial subgroups to
the exposure of PUAs (Figure 8B). The main contributing genus for the promotion effect by high-dose
PUAs was the group of *Alteromonas* spp., which showed a large increase in abundance by 73.9% and
69.7% in the middle and the bottom layers. For low-dose PUAs, the promotion effect of PUAs on
*Alteromonas* spp. was still found although with a much lower intensity (5.4% in the middle and 19.4% in
the bottom). The promotion effect of γ-Pro by high-dose PUAs was also contributed by bacteria *Halomonas*
spp. (percentage increase from 1.7% to 7.4%). Meanwhile, some bacterial genus, such as *Marinobacter* and
*Methylophaga* from γ-Pro, or *Nautella* and *Sulfitobacter* from α-Pro, showed decreased percentages by
high-dose PUAs (Figure 8B).

**3. 5 Carbon source preclusion experiments for PUAs**

After one month of incubation, PAB inoculated from the low-oxygen waters showed dramatic responses to both PAH and ALK (Figure 9). In particular, the mediums of PAH addition became turbid brown (bottles on the left) with the medium of ALK addition turning into milky white (bottles in the middle) (Figures 9B and 9D). For comparison, they were both clear and transparent at the beginning of the experiments (Figures 9A and 9C). These results should reflect the growth of bacteria in these bottles with the enrichments of organic carbons. Meanwhile, the minimal medium with the addition of heptadienal (C7_PUA) remained clear and transparent as it was originally, which would indicate that PAB did not grow in the treatment of C7_PUA.

**4. Discussion**

Hypoxia occurs if the rate of oxygen consumption exceeds that of oxygen replenishment by diffusion, mixing, and advection (Rabouille et al., 2008). The spatial mismatch between the surface chlorophyll-*a* maxima and the subsurface hypoxia during our estuary-to-shelf transect should indicate that the low-oxygen feature may not be directly connected to particle export by the surface phytoplankton bloom. This outcome can be a combined result of riverine nutrient input in the surface, water-column stability driven by wind and buoyancy forcing, and flow convergence for an accumulation of organic matters in the bottom (Lu et al., 2018).

Elevated concentrations of pPUAs and dPUAs near the bottom boundary of the salt-wedge should reflect a sediment source of PUAs, as the surface phytoplankton above them was very low. PUA-precursors such as PUFA could be accumulated as detritus in the surface sediment near the PRE mouth during the spring blooms (Hu et al., 2006). Strong convergence at the bottom of the salt-wedge could be driven by shear vorticity and topography (Lu et al., 2018). This would allow for the resuspension of small detrital particles. Improved PUAs production by oxidation of the resuspended PUFA could occur below the salt-wedge as a result of enhanced lipoxygenase activity (in the resuspended organic detritus) in response to salinity increase by the intruded bottom seawater (Galeron et al., 2018).

Direct measurement of the adsorbed PUAs concentration associated with the suspended particles
of >25 μm by the method of combined large-volume filtration and on-site derivation and extraction yield a
high level of ~10 μmol $L^{-1}$ within the hypoxic zone. This value is comparable to those previously reported
in sinking particles (>50 μm) of the open ocean using particle-volume calculated from diatom-derived
marine snow particles (Edward et al., 2015). Note that there was also a higher level of 240 μmol $L^{-1}$ found
in another station outside the PRE. A micromolar level of particle-adsorbed PUAs could act as a hotspot for
bacteria likely exerting important impacts as signaling molecules on microbial utilization of particulate
organic matters and subsequent oxygen consumption.
It should be mentioned that various pore sizes have been used for separating PAB sampling in the
literature. A 0.8-μm filtration was generally accepted for separating PAB (>0.8 μm) and FLB (0.2-0.8 μm)
in the ocean (Robinson and Williams, 2005; Kirchman, 2008; Huang et al., 2018; Liu et al., 2020). Other
studies defined size of >3 μm for PAB and 0.2-3 μm for FLB in some coastal waters (Crump et al. 1999;
Garneau et al., 2009; Zhang et al., 2016). Meanwhile, there were also many studies using much larger sizes
of filtration for PAB: a 5-μm filter in the German Wadden Sea (Rink et al. 2003), a 10-μm filter in the Santa
Barbara Channel (DeLong et al. 1993), a 30-μm filter in the Black Sea (Fuchsman et al., 2011), and a
50-μm-mesh nylon net in the North Atlantic waters (Edwards et al., 2015).
The hypoxic waters below the salt-wedge have high turbidity probably due to particle resuspension.
High particle concentration here may explain the pervious finding of a higher abundance of PAB than FLB
in the same area (e.g. Li et al., 2018; Liu et al., 2020), similar to those found in the Columbia River estuary
(Crump et al., 1998). Also, anaerobic bacteria and taxa preferring low-oxygen conditions were found more
enriched in the particle-attached communities than their free-living counterparts in the PRE (Zhang et al.,
2016). Our field measurements suggested that bacterial respiration within the hypoxic waters was largely
contributed by PAB (>0.8 μm) with FLB (0.2-0.8 μm) playing a relatively small role. Therefore, it is
crucial to address the linkage between the high-density PAB and the high level of particle-adsorbed PUAs
associated with the suspended particles in the low-oxygen waters.
We choose a larger pore-size of 25 μm for collecting bacteria attached to sinking aggregates and large
suspended particles. Firstly, it has been suggested that microbial respiration rate can be positively related to
aggregate size (Ploug et a., 2002) and thus larger PAB likely contributes more to oxygen consumption.
Secondly, larger particle size can better present the PAB taxonomy according to the previous finding of the
saturation of species-accumulation (for the size-fractionated bacteria) when the size is greater than 20 μm
(Mestre et al., 2017). Thus, the taxonomic groups of PAB caught on particles of >25 μm should already
cover those of PAB on smaller particles of 0.8-25 μm. A similar type of filtration (30 μm) has been
previously applied to study PAB in the Black Sea suboxic zones (Fuchsman et al., 2011).
Interestingly, our PUA-amended experiments for PAB (>25 μm) retrieved from the low-oxygen waters
revealed distinct responses of PAB to different doses of PUAs treatments with an increase in cell growth in
response to low-dose PUAs (1 μmol L$^{-1}$) but an elevated cell-specific metabolic activity including bacterial
respiration and production in response to high-dose PUAs (100 μmol L$^{-1}$). An increase in cell density of
PAB by low-dose PUAs could likely reflect the stimulating effect of PUAs on PAB growth. This finding
was consistent with the previous report of a PUAs level of 0-10 μmol L$^{-1}$ stimulating respiration and cell
growth of PAB in sinking particles of the open ocean (Edwards et al., 2015). The negligible effect of
low-dose PUAs on bacterial community structure in our experiments was also in good agreement with
those found for PAB from sinking particles (Edwards et al., 2015). However, we do not see the inhibitory
effect of 100 μmol L$^{-1}$ PUAs on PAB respiration and production previously found in the open ocean
(Edward et al., 2015). Instead, the stimulating effect for high-dose PUAs on bacterial respiration and
production was even stronger with ~50% increments. The bioactivity of PUAs on bacterial strains could
likely arise from its specific arrangement of two double bonds and carbonyl chain (Ribalet et al., 2008).
Our findings support the important role of PUAs in enhancing bacterial oxygen utilization in low-oxygen
waters.
It should be mentioned that it remains controversial the effect of background nanomolar PUAs on
free-living bacteria, which is not our focus in this study. Previous studies suggested that 7.5 nmol L$^{-1}$ PUAs
would have a different effect on the metabolic activities of distinct bacterial groups in the NW
Mediterranean Sea, although bulk bacterial abundance remained unchanged (Balestra et al., 2011). In
particular, the metabolic activity of γ-Pro was least affected by nanomolar PUAs, although those of
Bacteroidetes and Rhodobacteraceae were markedly depressed (Balestra et al., 2011). Meanwhile, the daily
addition of 1 nmol L$^{-1}$ PUAs was found to not affect the bacterial abundance and community composition
during a mesocosm experiment in the Bothnian Sea (Paul et al., 2012).

488        It is important to verify that the PUAs are not an organic carbon source but a stimulator for PAB

growth and metabolism. This was supported by the fact that the inoculated PAB could not grow in the
medium with 200 μmol L$^{-1}$ of PUAs although they grew pretty well in the mediums with a similar amount
of ALK or PAH. Our results support the previous findings that the density of *Alteromonas hispanica* was
not significantly affected by 100 μmol L$^{-1}$ of PUAs in the minimal medium (without any organic carbons)
during laboratory experiments (Figure 9E), where PUAs were considered to act as cofactors for bacterial
growth (Ribalet et al., 2008).

495        Improved cell-specific metabolism of PAB in response to high-dose PUAs was accompanied by a

significant shift of bacterial community structure. The group of PAB with the greatest positive responses to
exogenous PUAs was genus *Alteromonas* within the γ-Pro, which is well-known to have a particle-attached
lifestyle with rapid growth response to organic matters (Ivars-Martinez et al., 2008). This result is
contradicted by the previous finding of a reduced percentage of the γ-Pro class by high-dose PUAs in the
PAB of open ocean sinking particles (Edward et al., 2015). Meanwhile, previous studies suggested that
different genus groups within the γ-Pro may respond distinctly to PUAs (Ribalet et al., 2008). Our result
was well consistent with the previous finding of the significant promotion effect of 13 or 106 μmol L$^{-1}$
PUAs on *Alteromonas hispanica* from the pure culture experiment (Ribalet et al., 2008). An increase of
PUAs could thus confer some of the γ-Pro (mainly special species within the genus *Alteromonas*, such as *A.*
*hispanica*, Figure S2B) a competitive advantage over other bacteria, leading to their population dominance
on particles in the low-oxygen waters. These results provide evidences for a previous hypothesis that PUAs
could shape the bacterioplankton community composition by driving the metabolic activity of bacteria with
neutral, positive, or negative responses (Balestra et al., 2011).
The taxonomic composition of PAB (>25 μm) was substantially different from that of the bulk
bacteria community in the hypoxic zone (with a large increase of γ-Pro associated with particles, Figure
S2A). This result supports the previous report of γ-Pro being the most dominant clades attached to sinking
particles in the ocean (DeLong et al., 1993). A broad range of species associated with γ-Pro was known to
be important for quorum sensing processes due to their high population density (Doberva et al., 2015)
associated with sinking or suspended aggregates (Krupke et al., 2016). In particular, the genus of γ-Pro such
as *Alteromonas* and *Pseudomonas*, are well-known quorum-sensing bacteria that can rely on diverse
signaling molecules to affect particle-associated bacterial communities by coordinating gene expression
within the bacterial populations (Long et al., 2003; Fletcher et al., 2007).
It has been reported that the growths of some bacterial strains of the γ-Pro such as *Alteromonas* spp.
and *Pseudomonas* spp. could be stimulated and regulated by oxylipins like PUAs (Ribalet et al., 2008; Pepi
et al., 2017). Oxylipins were found to promote biofilm formation of *Pseudomonas* spp. (Martinez et al.,
2016) and could serve as signaling molecules mediating cell-to-cell communication of *Pseudomonas* spp.
by an oxylipin-dependent quorum sensing system (Martinez et al., 2019). As PUAs are an important group
of chemical cues belonging to oxylipins (Franzè et al., 2018), it is thus reasonable to expect that PUAs may
also participate as potential signaling molecules for the quorum sensing among a high-density *Alteromonas*
or *Pseudomonas*. A high level of particle-adsorbed PUAs occurring on organic particles in the low-oxygen
water would likely allow particle specialists such as *Alteromonas* to regulate bacterial community structure,
which could alter species richness and diversity of PAB as well as their metabolic functions such as
respiration and production when interacting with particulate organic matter in the hypoxic zone. Various
bacterial assemblages may have different rates and efficiencies of particulate organic matter degradation
(Ebrahimi et al., 2019). Coordination amongst these PAB could be critical in their ability to thrive on the
recycling of POC (Krupke et al., 2016) and thus likely contribute to the acceleration of oxygen utilizations
in the hypoxic zone. Nevertheless, the molecular mechanism of the potential PUA-dependent quorum
sensing of PAB may be an important topic for future study.

534        Our findings may likely apply to other coastal systems where there are large river inputs, intense

phytoplankton blooms driven by eutrophication, and strong hypoxia, such as the Chesapeake Bay, the
Adriatic Sea, and the Baltic Sea. For example, Chesapeake Bay is largely influenced by river runoff with
strong eutrophication-driven hypoxia during the summer as a result of increased water stratification (Fennel
and Testa, 2019) and enhanced microbial respiration fueled by organic carbons produced during spring
diatom blooms (Harding et al., 2015). Similar to the PRE, there was also a high abundance of $\gamma$-Pro in the
low-oxygen waters of the Chesapeake Bay associated with the respiration of resuspended organic carbon
(Crump et al., 2007). Eutrophication causes intense phytoplankton blooms in the coastal ocean.
Sedimentation of the phytoplankton carbons will lead to their accumulation in the surficial sediment
(Cloern, 2001), including PUFA compounds derived from the lipid production. Resuspension and oxidation
of these PUFA-rich organic particles during summer salt-wedge intrusion might lead to high
particle-adsorbed PUAs in the water column. These PUAs could likely shift the particle-attached bacterial
community to consume more oxygen when degrading particulate organic matter and thus potentially
contribute to the formation of seasonal hypoxia. In this sense, the possible role of PUAs on coastal hypoxia
may be a byproduct of eutrophication driven by anthropogenic nutrient loading. Further studies are required
to quantify the contributions from PUAs-mediated oxygen loss by aerobic respiration to total
deoxygenation in the coastal ocean.

**5. Conclusions**
In summary, we found elevated concentrations of pPUAs and dPUAs in the hypoxic waters below the
salt-wedge. We also found high particle-adsorbed PUAs associated with particles of >25 $\mu$m in the hypoxic
waters based on the large-volume filtration method, which could generate a hotspot PUAs concentration
of >10 $\mu$mol L$^{-1}$ in the water column. In the hypoxic waters, bacterial respiration was largely controlled by
PAB (>0.8 μm) with FLB (0.2-0.8 μm) only accounting for 25-30% of the total respiration. Field
PUA-amended experiments were conducted for PAB associated with particles of >25 μm retrieved from the
low-oxygen waters. We found distinct responses of PAB (>25 μm) to different doses of PUAs treatments
with an increase of cell growth in response to low-dose PUAs (1 μmol $L^{-1}$) but an elevated cell-specific
metabolic activity including bacterial respiration and production in response to high-dose PUAs (100 μmol
$L^{-1}$). Improved cell-specific metabolism of PAB in response to high-dose PUAs was also accompanied by a
substantial shift of bacterial community structure with increased dominance of genus *Alteromonas* within
the γ-Pro.
Based on these observations, we hypothesize that PUAs may potentially act as signaling molecules for
coordination among the high-density PAB below the salt-wedge, which would likely allow bacteria such as
*Alteromonas* to thrive in degrading particulate organic matters. Very possibly, this process by changing
community compositions and metabolic rates of PAB would lead to an increase of microbial oxygen
utilization that might eventually contribute to the formation of coastal hypoxia.

*Data availability*. Some of the data used in the present study are available in the Supplement. Other data
analyzed in this article are tabulated herein. For any additional data please request from the corresponding
author.

*Supplement*. The supplement related to this article is available online at: bg-2020-243-supplement.

*Author Contributions*. Q.P.L designed the project. Z.W. performed the experiments. Q.P.L and Z.W. wrote
the paper with inputs from all co-authors. All authors have given approval to the final version of the
manuscript.

*Competing interests*. The authors declare no competing financial interest.

*Acknowledgements.* We are grateful to the captains and the staff of *R/V Haike68* and *R/V Tan Kah Kee* for

help during the cruises. We thank Profs Dongxiao Wang (SCSIO) and Xin Liu (XMU) for organizing the

cruises, Mr. Yuchen Zhang (XMU) for field assistances, Profs Changsheng Zhang (SCSIO) and Weimin

Zhang (GIM) for analytical assistance, as well as Prof. Dennis Hansell (RSMAS) for critical comments.

587

*Financial support.* This work was supported by the National Natural Science Foundation of China

(41706181, 41676108), the National Key Research and Development Program of China

(2016YFA0601203), and the Key Special Project for Introduced Talents Team of Southern Marine Science

and Engineering Guangdong Laboratory (Guangzhou) (GML2019ZD0305). ZW also wants to

acknowledge a visiting fellowship (MELRS1936) from the State of Key Laboratory of Marine

Environmental Science (Xiamen University).

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

**Table 1**. Summary of treatments in the experiments of exogenous PUAs additions for the low-oxygen
waters at station Y1 during June 2019. The PUAs solution includes heptadienal (C7_PUA), octadienal
(C8_PUA), and decadienal (C10_PUA) with the mole ratios of 10:1:10.

| | | Treatment |
|---|---|---|
| 1 | Control (methanol) | methanol |
| 2 | Low-dose PUAs (methanol) | 2 mM PUAs in methanol |
| 3 | High-dose PUAs (methanol) | 200 mM PUAs in methanol |



**Figure 1:** Sampling map of the Pearl River Estuary and the adjacent northern South China Sea during (A)
June 17$^{th}$-28$^{th}$, 2016, (B) June 18$^{st}$-June 2$^{nd}$, 2019. Contour shows the bottom oxygen distribution with
white lines highlighting the levels of 93.5 µmol kg$^{-1}$ (oxygen-deficient zone) and 62.5 µmol kg$^{-1}$ (hypoxic
zone); dashed line in panel A is an estuary-to-shelf transect with blue dots for three stations with bacterial
metabolic rate measurements; diamonds in panel B are two stations with vertical pPUAs and dPUAs
measurements with Y1 the station for PUAs-amended experiments.

**Figure 2:** Procedure of large-volume filtration and subsequent experiments. A large volume of the
low-oxygen water was filtered through a 25-µm filter to obtain the particles-adsorbed PUAs and the
particle-attached bacteria (PAB). The carbon-source test of PUA for the inoculated PAB includes the
additions of PUA, alkanes (ALK), and polycyclic aromatic hydrocarbons (PAH). PUAs-amended
experiments for PAB include Control (CT), Low-dose (PL), and High-dose PUAs (PH). Samples in the
biological oxygen demand (BOD) bottles at the end of the experiment were analyses for bacterial
respiration (BR), abundances (BA), production (BP) as well as DNA. Note that pPUAs and dPUAs are
particulate and dissolved PUAs in the seawater.

**Figure 3:** Vertical distributions of (A) temperature, (B) turbidity, (C) nitrate, (D) salinity, (E) dissolved
oxygen, and (F) chlorophyll-*a* from the estuary to the shelf of the NSCS during June 2016. Section
locations are shown in Figure 1; the white line in panel D shows the area of oxygen deficiency zone (<93.5
µmol kg$^{-1}$).

**Figure 4:** Comparisons of oxygen, bulk bacterial respiration (BR) and production (BP), as well as bulk
bacterial abundances (BA) of α-Proteobacteria (α-Pro), γ-Proteobacteria (γ-Pro), Bacteroidetes (Bact), and
other bacteria for the bottom waters between stations inside (X1) and outside (X2 and X3) the hypoxic zone
during the 2016 cruise. Bulk bacteria community includes FLB and PAB of <20 µm. Locations of stations
X1, X2, X3 are showed in Figure 1A. Error bars are the standard deviations.

**Figure 5:** Vertical distributions of (A) temperature, (B) salinity, (C) dissolved oxygen (DO), (D)
chlorophyll-*a* (Chl-*a*), (E) particulate PUAs (pPUAs) and (F) dissolved PUAs (dPUAs) inside (Y1) and
outside (Y2) the hypoxic zone during June 2019. Locations of station Y1 and Y2 are shown in Figure 1.
Error bars are the standard deviations.

**Figure 6:** Concentrations of particle-adsorbed PUAs (in micromoles per liter particle) in the middle (12 m) and the bottom (25 m) waters of station Y1 during June 2019. Three different PUA components are also shown including heptadienal (C7_PUA), octadienal (C8_PUA), and decadienal (C10_PUA). Error bars are the standard deviations.

**Figure 7:** Responses of particle-attached bacterial parameters including (A) bacterial abundance (BA$_{particle}$), (B) bacterial respiration (BR$_{particle}$), (C) cell-specific bacterial respiration (sBR$_{particle}$), (D) bacterial growth efficiency (BGE$_{particle}$), (E) bacterial production (BP$_{particle}$), and (F) cell-specific bacterial production (sBP$_{particle}$) to different doses of PUAs additions at the end of the experiments for the middle (12 m) and the bottom waters (25 m) at station Y1. Error bars are standard deviations. The star represents a significant difference ($p<0.05$) with PL and PH the low and high dose PUA treatments and C the control.

**Figure 8:** Variation of particle-attached bacterial community compositions on (A) the phylum level and (B) the genus level in response to different doses of PUAs additions at the end of the experiments for the middle and the bottom waters at station Y1. Labels PL and PH are for the low- and high-dose PUAs with CT the control.

**Figure 9:** Carbon-source test of PUAs with cell culture of particle-attached bacteria inoculated from the low-oxygen waters of station Y1 including the initial conditions (Day0) at the beginning of the experiments as well as results after 30 days of incubations (Day30) for (A, B) the middle and (C, D) the bottom waters, respectively. Bottles from left to right are the mediums (M) with the additions of polycyclic aromatic hydrocarbons (M+PAH, 200 ppm), alkanes (M+ALK, 0.25 g L$^{-1}$), and heptadienal (M+C7_PUA, 0.2 mmol L$^{-1}$); Note that a change of turbidity should indicate bacterial utilization of organic carbons. (E) the optical density of bacterium *Alteromonas hispanica* MOLA151 growing in the minimal medium as well as in the mediums with the additions of mannitol, pyruvate, and proline (M+MPP, 1% each,), heptadienal (M+C7_PUA, 145μM), octadienal (M+C8_PUA, 130μM,), and decadienal (M+C10_PUA, 106μM). The method for *A. hispanica* growth and the data in panel E are from Ribalet et al., 2008.


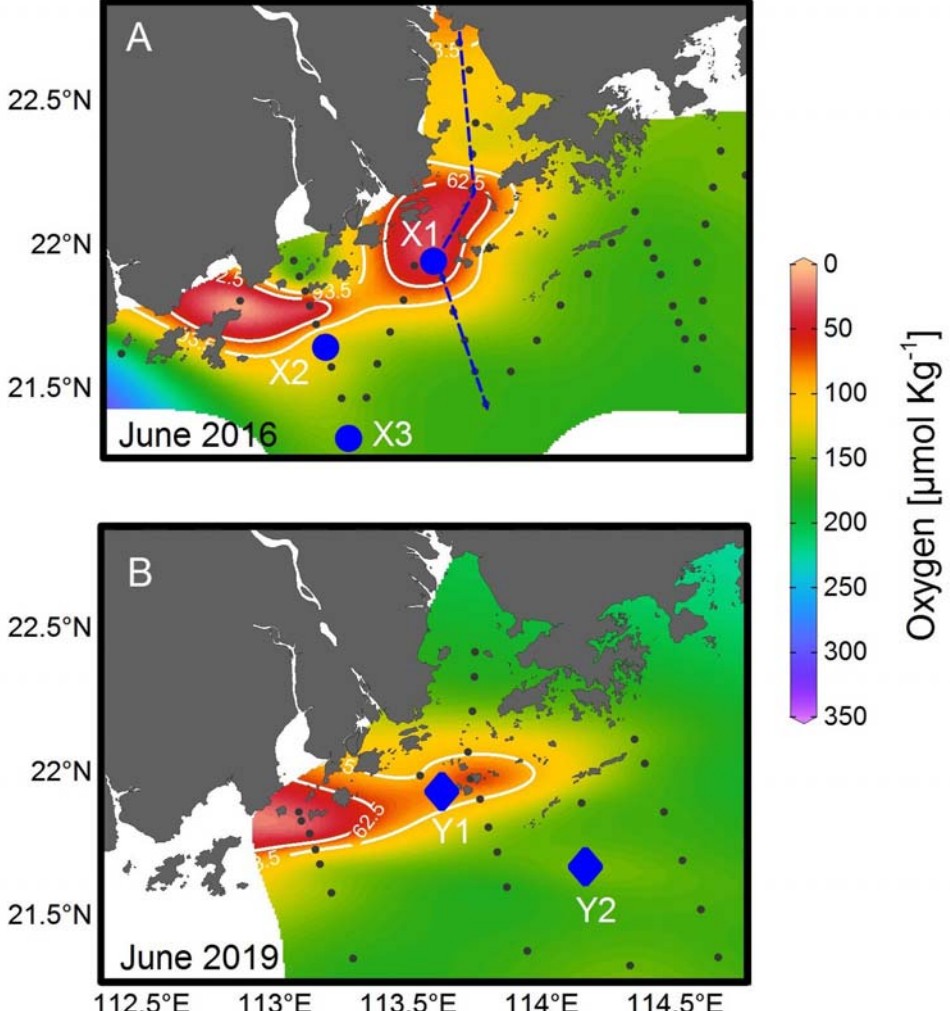



836                    **Figure 1**


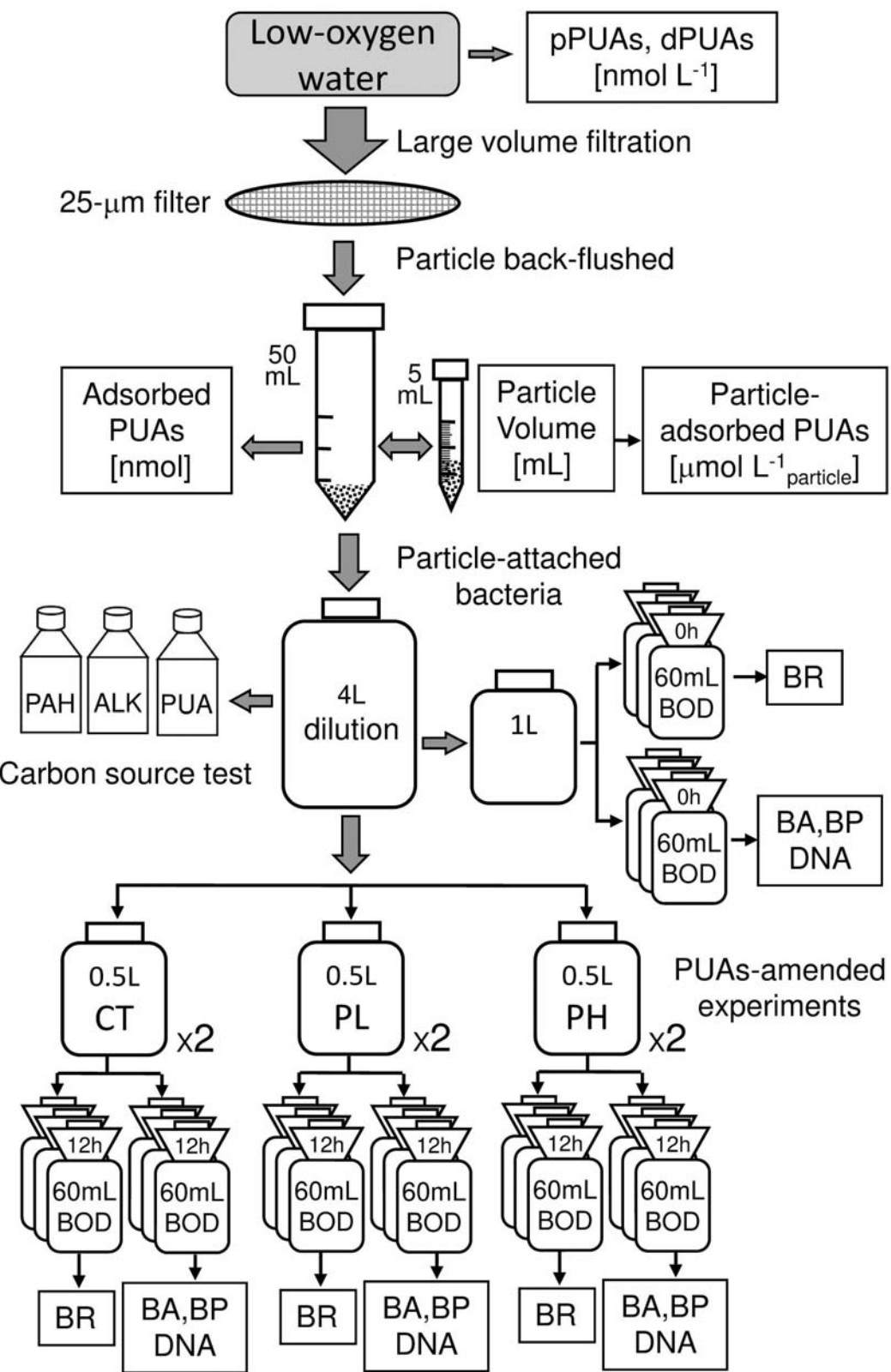


**Figure 2**

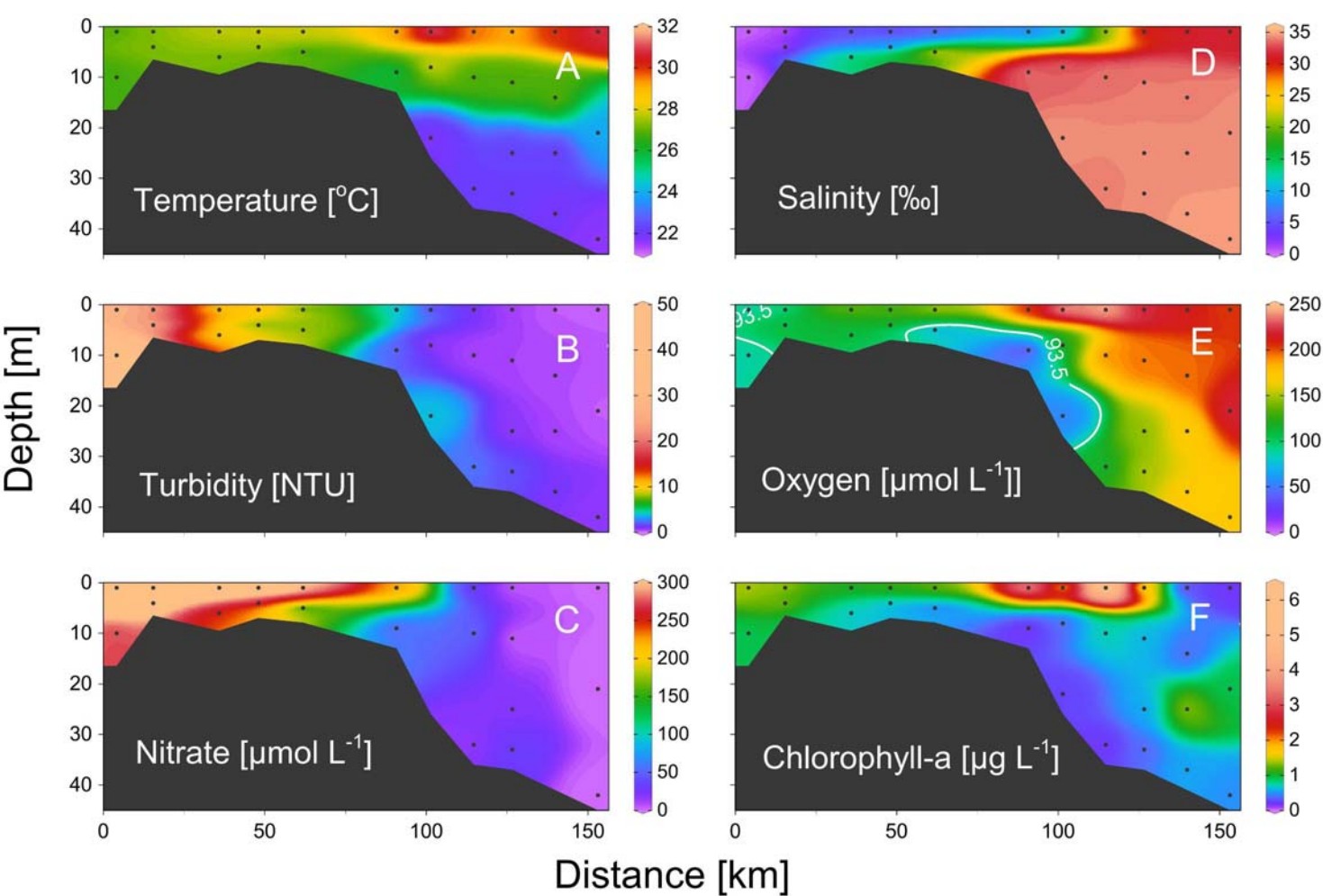

**Figure 3**

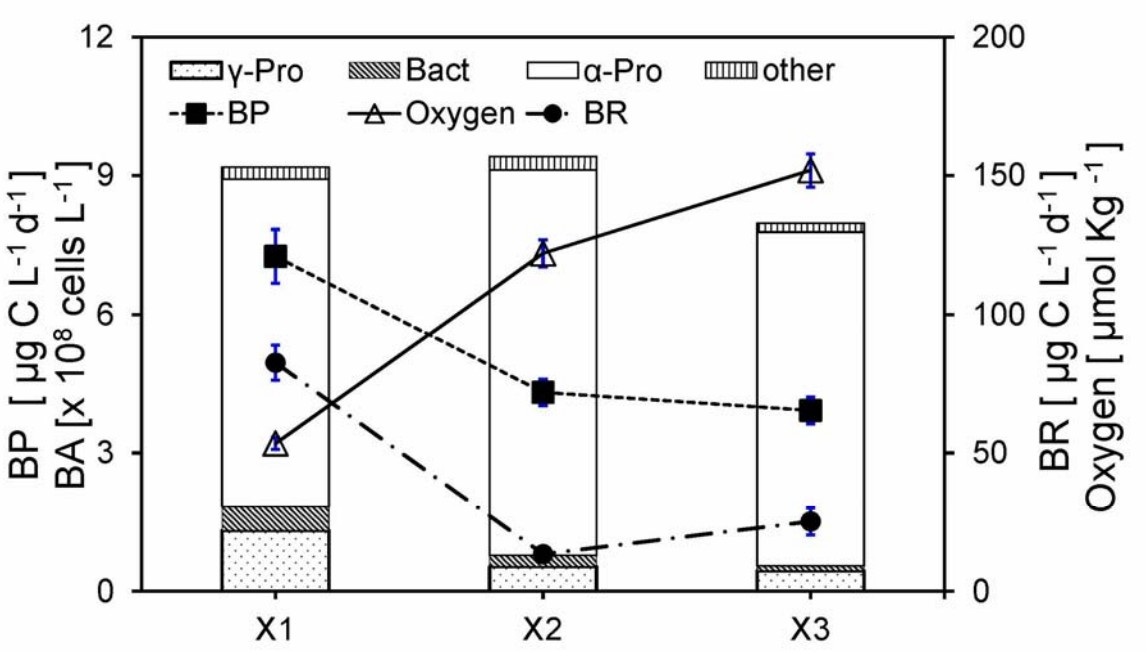



**Figure 4**

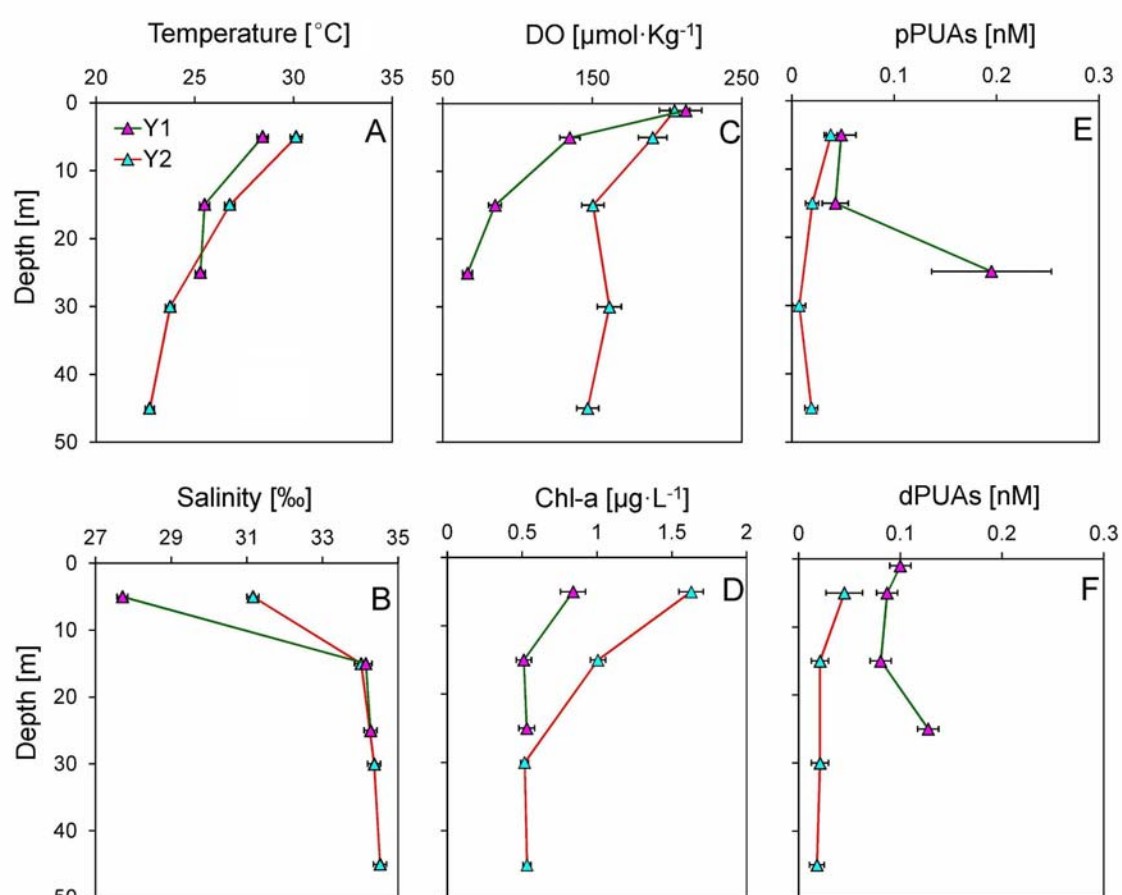

**Figure 5**

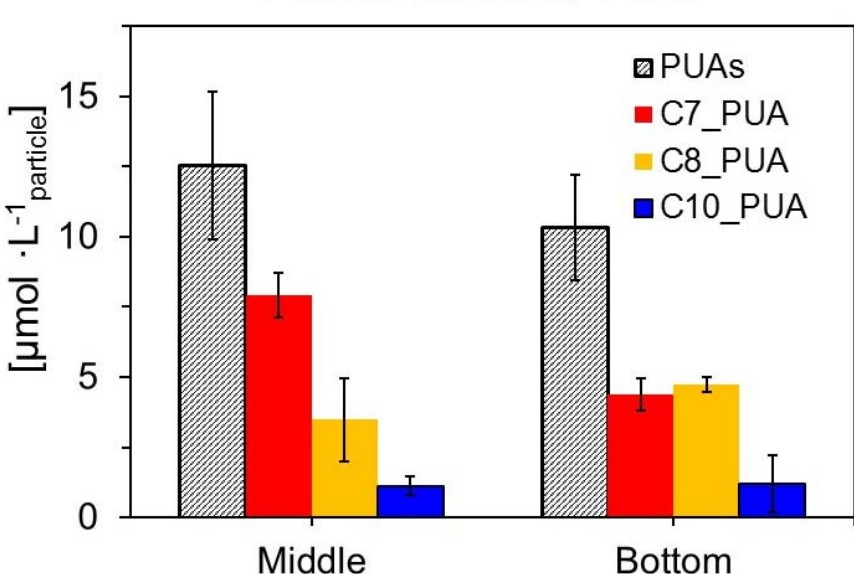

**Figure 6**





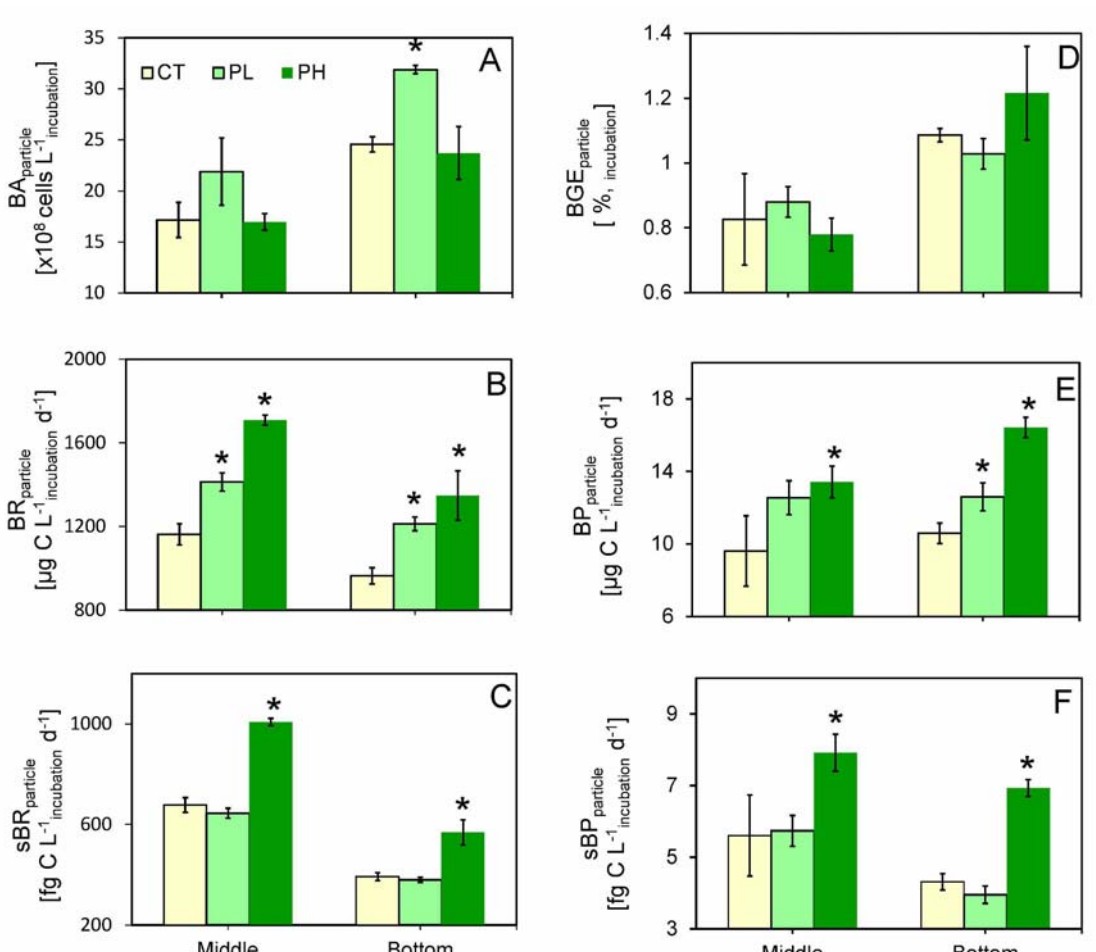



**Figure 7**

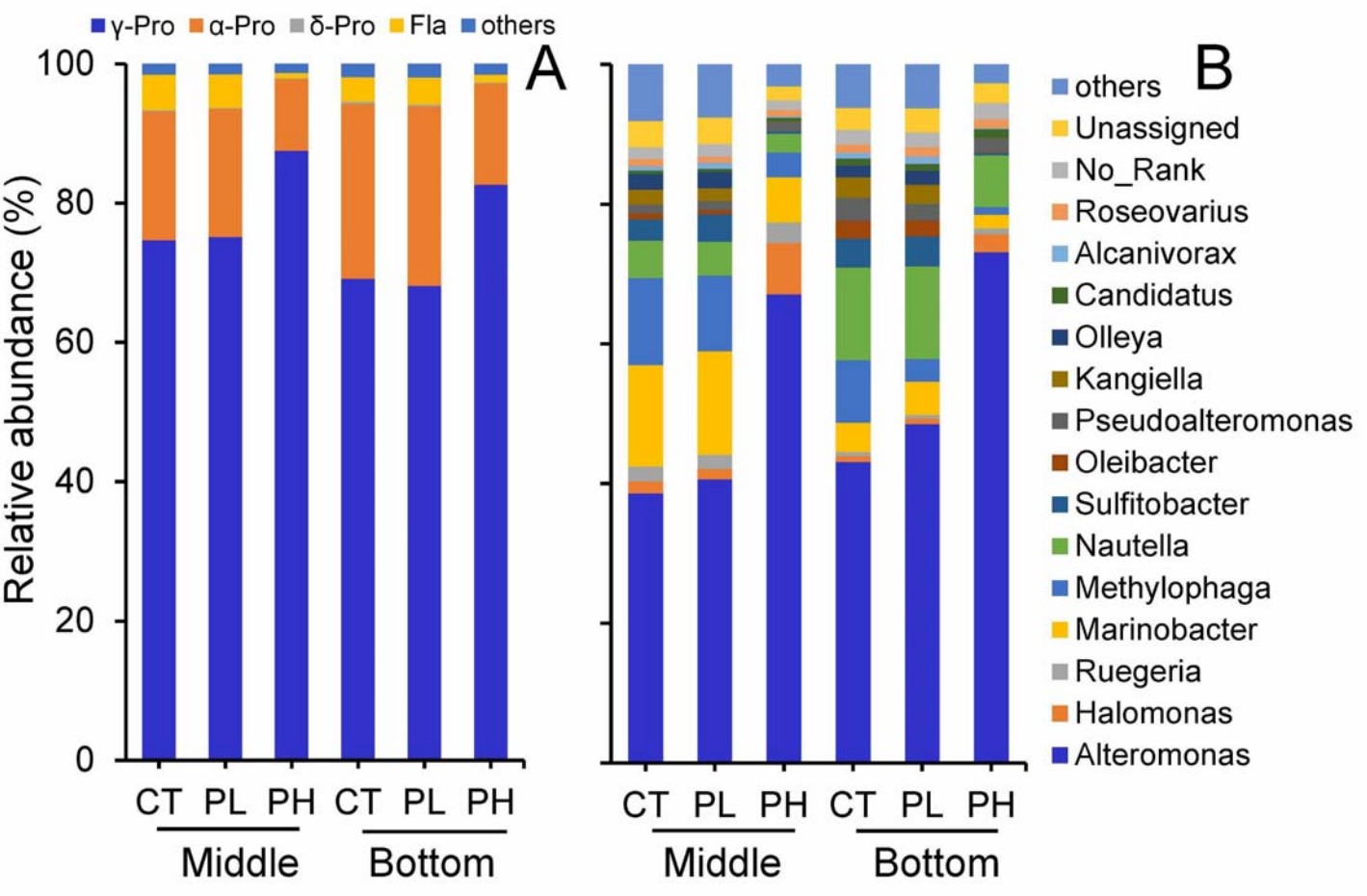



**Figure 8**

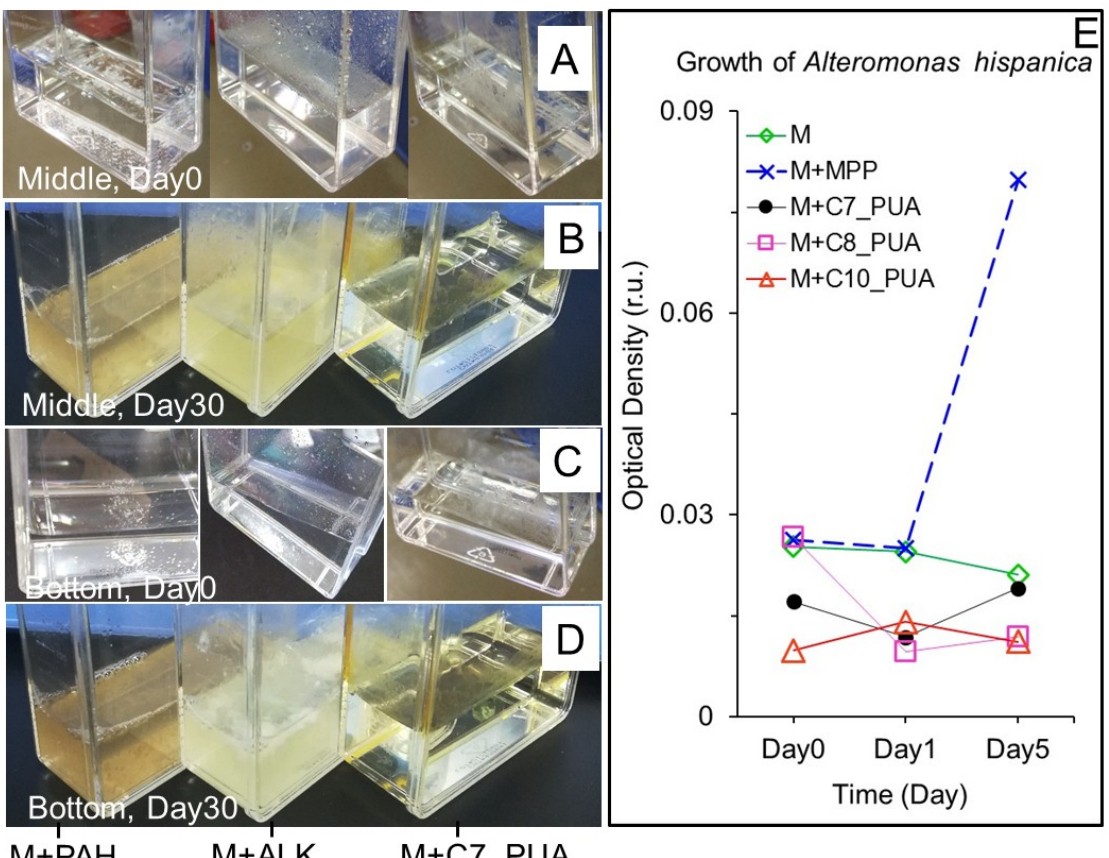

M+PAH          M+ALK          M+C7_PUA

**Figure 9**