# Peer review of "Impacts of biogenic polyunsaturated aldehydes on metabolism and community"

_Biogeosciences, 2020_

## Referee Comment (RC1) · Anonymous Referee #1 · 9 Sep 2020

Review Wu et al 2020, bg-2020-243 Title_ Impacts of biogenic polyunsaturated aldehydes on metabolism and community composition of particle-attached bacteria in coastal hypoxia. General comments The authors study poly-unsaturated aldehydes (PUA) and their influence on particle-attached bacterial abundance, taxonomy and activity. This is done in an partially hypoxic coastal area of the Pearl River Estuary during June in two consecutive years using 12 h to 30 d incubation. The authors conclude that PUA can influence bacterial abundance, taxonomy, growth and respiration. The manuscript is generally well written with proper language although some sentences need improve-

ments. Results are well presented in the figures, but some complement of error bars are needed. I lack a convincing motivation to the importance of PUAs compared to the multitude of other organic compounds, and the importance of particle-attached bacteria as compared to free-living. No direct comparison with other organic compounds or free-living bacteria is done in the study. A general importance of PUAs and particle-attached bacteria should be tuned down in the discussion and conclusion. The effects of PUA on particle-attached bacteria is still of value as such. A short-coming of the experimental design is a lack of true replication of the treatments. In addition, the fact that only one season has been investigated. I also miss proper measurement of the abundance, acidity and taxonomy of free-living bacterial to put the claimed influence of particle attached bacteria in perspective. A similar argument for the lack of other organic compounds in the study. The conclusions must therefore be made more cautious, specific and these shortcomings commented on. Some speculative statements in the discussion and conclusion section need to be removed or rephrased. There are parts of the method descriptions that need to be clarified, better specified or added. A major revision in this spirit is required to motivate publication. Detailed comments r. 13-14 There is an extensive literature on eutrophication driven hypoxia in e.g. the Baltic Sea since 4 decades (cf. Cloern 2001). Please rephrase sentence accordingly r. 15. Do you mean "...water mainly dominated...". r. 21 Please change "activity" to "..e.g. bacterial respiration and growth...) and revise the sentence. r. 35 Change to "..deoxygenation is also tightly..." r. 43-45 In most aquatic environments free living bacteria are dominating in numbers as well as biomass (e.g. Kirchman 2008). The reference is not convincingly showing that particle-associated bacterial dominate in terms of abundance or how the growth of particle associated bacteria was measured. Please revise the message. This also question the focus on particle associated bacteria in the manuscript. r. 49-51 In many cases free-living bacteria dominate the respiration (Robinson and leB Williams 2005). Both types of bacteria is preferably studied. This question the general relevance of the study. r. 64-65. However, many other organic compounds may drive the bacterial respiration. Please provide some reference showing to that extent PUF is contributing to bacterial respiration. r.68-76 I would prefer more explicit research questions to be addressed in this paragraph for clarity and coupling to performed experiments. r. 84-85 Please define here what depths that were used for middle and bottom water categories (e.g. figure 6). r. 87-88 Please define the abbreviations pPUA and dPUA. r. 97 Filtration and freezing of nutrient samples may release nutrients from broken cells. r.124-129 How is the centrifugation and re-suspension of particles influencing their morphology, attachments of PUAs and PABs? r. 131-132 Pleas provide the recovery efficiency of particle attached PUAs after the preparation procedure described. r. 136-137. Please provide a reference where the method is validated. r. 146-147 Freeze thawing may relate PUAs from living cells also. r. 152 Should it be nmol per some volume or particle unit? r.160 Please specify what is meant by clean. What was the washing procedure? r.162 As presented here there was no true replication of the treatments? r. 167 How does methanol included in the procedure affect bacterial abundance and activity? Any control or test for this? Please comment on relevance for natural conditions. r. 176-179 Give some information on how close to natural conditions these final concentration of PUAs are. r.180 it is not obvious that turbidity will be detected if cells remain below about 109 cells cm3. Were the cell concentration measured by direct microscopy? r. 181 Please provide a description on incubation conditions and length. r. 185-196 The description of methodology is unclear. Please make clear if and how free-living bacterial abundance was measured? How is the methanol treatment accounting for bacteria from breaking particles? IN addition, please provide a reference validating this method. Give some measure of the precision of the flow cytometer analysis and a relevant reference. r.198-202 As no relevant pre-filtration is used you may include other organisms than bacteria in the respiration estimate . Please clarify and rephrase as needed. r. 208-209 What was the final TCS concentration in the sample. This does not follow the common procedure. Neither use of ethanol. r. 241 However, the lack of true replicates for the treatment (i.e. replicate 1-L Nalgene bottles per treatment) question a reliable result from the t-test. r. 251, Figure 1. Consider to reverse the colour palette. More logical to have blue for well oxygenated

and red for hypoxia. Change "constant" to "similar". r. 261-263. Pleas provide a statistical test for the claimed difference and confidence intervals (error bars) in figure 4. r.265. Proved statistical results for the claimed difference between bacterial phyla. Do the same for other differences claimed throughout the manuscript. r. 273 Should it be Bacteroidetes also here? Figure 5. Please present the type of error bars used. Same for all figures with error bars. r. 320-329 What can be considered significant differences as opposed to random variation in this analysis. Please motivate convincingly. r. 335 One month is an extremely long incubation. How relevant is this for the application to the natural environment? r. 360 Wold be more informative to use a unit per particle or mass of particles? Litre of particles is unclear. r. 365 This assumes that PUA is a major substrate among all other organic compounds. Please provide some references on this matter and discuss it critically. r. 368 This should be compared with the biomass of free living bacteria. They may also be elevated in the hypoxic water. I find the lack o measurement of free-living bacteria a short coming in the context of claiming importance of PABs. r. 372-373. How is respiration by particle-attached bacteria distinguished from protozoa, phytoplankton and larger zooplankton? This is typically difficult to achieve. Comment in a critical manner. r. 376-390. Given the apparent lack of true replication of the treatments (i.e. replicate 1 L Nalgene bottles) the conclusions regarding treatment effects is highly uncertain. This needs a discussion. r. 389-390 Relevance in the natural environment assumes that the applied concentrations are relevant for those occurring in the natural environment. Please consider, discuss and modify the conclusion accordingly. r. 391-392 I find it valuable to know if PUA stimulates bacterial activity whether as an organic substrate of metabolic signal substance. Please explain why only the latter would be ecologically important. r. 392-394 Please use the same concentration unit for comparability of levels. Heptadienal alone used for the test may not be comparable to a mixture of different PUAs (i.e. concentration more than twice used in the combined concentration). Why was not the same mixture used for this experiment? Other methods like using labelled PUA and analyse for metabolism of those would better test the mechanism of PUA effect. r. 405-407 Please consider that a few

species within the Alteromonas phylum may be responsible for the observed response. PUA metabolism might not be a function attributed to the whole phyla. Rephrase the discussion accordingly. r. 411-412 Please refer to what figure and test that show a difference between particle-attached and bulk bacteria. r. 427 How strongly are PUAs adsorbed to particles (i.e. chemical bonding)? How may this influence their potential to act as signalling molecules? r. 439-441 I have not seen any analyses of the lipoxygenase hypothesis in the study? It is thus speculative and should be removed. Focus on conclusion that can be derived from the performed study. r. 442-445 As there was no true replicates this conclusion should be made more cautious. r.455-460 If this section should remain it need to be moved to the discussion section. r. 460-464 The sudden appearance of PUFA is not connected to the previous sentence?. Again, this part is highly speculative and not part of conclusion from the study. Remove or move parts to the discussion. Literature cited Cloern, J. E. 2001. Our evolving conceptual model of the coastal eutrophication problem. Mar. Ecol.-Prog. Ser. 210: 223-253, doi. Robinson, C., and P. J. Le B Williams. 2005. Respiration and it′s measurement in surface marine waters, p. 147-180. In P. A. del Giorgio and P. J. Williams, le B [eds.], Respiration in aquatic ecosystems. Oxford University Press.

---

## Referee Comment (RC2) · Anonymous Referee #2 · 22 Sep 2020

Wu et al., reported impacts of biogenic polyunsaturated aldehydes on metabolism and community composition of particle-attached bacteria in coastal hypoxia. I think the manuscript is interesting, in particularly aiming at providing a biological mechanism underlying estuarine seasonal hypoxia. However, there appeared to be some problems about the experimental design of this study and the manuscript fails to provide convincing results. More details about the motivation and experiment procedure should be included and clarified and the conclusions should be carefully justified. 1. My major concerns about this manuscript is that the authors consider PAB as bacteria attached

to particles with a size >25 $\mu$m in the microcosm incubation. A great variety of bacteria would be lost, which would affect the major conclusion of the manuscript. The abundance of bacteria attached to >25-$\mu$m particles would be significantly lower than that of free-living bacteria. Also, is this particle size proper for measurement of polyunsaturated aldehydes? In addition, it is not clear why the authors choose 1 or 100 $\mu$mol L-1 but not the background value for the incubation. Moreover, bacterial community of the initial inoculates was lacking. Personally, the most interesting part of this study is the role of polyunsaturated aldehydes-enhanced bacterial oxygen demand for the seasonal hypoxia. Thus, it is important to know to what extent different concentration of polyunsaturated aldehydes affect bacterial growth and respiration. Although the authors provide discussion on this, more analyses including the selection of background concentration of polyunsaturated aldehydes and testing on pure isolates are needed. 2. It is unclear why the authors studied the effect of polyunsaturated aldehydes of bacterial communities. The importance of polyunsaturated aldehydes was not properly and clearly presented. For examples, although the authors mentioned the effect of polyunsaturated aldehyde on marine microorganisms, detailed processes and mechanisms are not provided. Line 43: Please provide reference for the higher abundance of PAB compared to FLB. Line 59-60: How polyunsaturated aldehyde affects marine microorganisms? Please provide more details. Line 63: What is the meaning of affect oxygen depletion? Is this a promoting or inhibiting process? Line 72-73: It is strange to place this sentence here. Why PUAs did not serve as carbon source? Line 81: Is that "July 2nd"? Line 88: what do you mean by "pPUAs and dPUAs"? Line 265: I did see description of methods about the bacterial community analysis in bottom waters of X1,X2, X3 and PAB on particles of >25 ïA■m. Line 276: Please provide the concentration of pPUAs and dPUAs. Line 278: shown Line 391: According to the results, low dose (1 $\mu$mol) of PUAs can stimulate the growth of PAB, significantly different from that of high dose (100 $\mu$mol) treatment. However, the test of PUAs as organic carbon source was conducted with 200 $\mu$mol of PUAs. I guess such a high concentration would adversely affect bacteria growth, while the low dose PUAs is likely to be used as organic sources.

Line 686: data of panel D are reproduced from Ribalet et al., 2008. Is this panel E? No methods were provided for growth of Alteromonas hispanica MOLA151. Line 448: Since bacteria on >25-$\mu$m particles can be low, hypothesis on signaling molecules may be tuned down.

---

## Author Comment (AC1) · 22 Oct 2020

General comments

1. The manuscript is generally well written with proper language although some sentences need improvements. Results are well presented in the figures, but some complement of error bars are needed.

Response: Thanks to the reviewer for constructive comments. The manuscript has been proofread by a native English speaker to correct grammar errors and to improve

the written language. We have redone these figures by including the error bars in the revised manuscript.

2. I lack a convincing motivation to the importance of PUAs compared to the multitude of other organic compounds, and the importance of particle-attached bacteria as compared to free-living. No direct comparison with other organic compounds or free-living bacteria is done in the study.

Response: The reviewer is right that there are other organic compounds that may also likely affect bacterial respiration, such as 2-n-pentyl-4-quinolinol (Long et al., 2003) and acylated homoserine lactones (Hmelo et al., 2011). Meanwhile, a perennial bloom of PUA-producing diatoms in the PRE mouth (Wu and Li, 2016) should argue for the importance of PUAs for microbial activity here compared to many other organic compounds. The reviewer is also right that free-living bacteria are important for community respiration in the ocean. However, our focus here is on the coastal transition zone, where particle-attached bacteria could be more important. In the revised manuscript, we have provided data of FLB respiration compared to the bulk bacterial community respiration in the hypoxic waters (FLB respiration account for only 25-30% of the total bacterial community).

3. A general importance of PUAs and particle attached bacteria should be tuned down in the discussion and conclusion. The effects of PUA on particle-attached bacteria is still of value as such.

Response: OK. We have rewritten the relevant sentences in the discussion and the conclusion sections as suggested by the reviewer.

4. A short-coming of the experimental design is a lack of true replication of the treatments.

Response: We actually have two replicates for each treatment. The original text was not well written. We have clarified this in the revised manuscript.

5. In addition, the fact that only one season has been investigated.

Response: The hypoxia could only occur during the summer in our study region. Therefore, a seasonality of hypoxia suggested by the reviewer may not be unnecessary.

6. I also miss proper measurement of the abundance, acidity and taxonomy of free-living bacterial to put the claimed influence of particle attached bacteria in perspective.

Response: We have added data of the size-fractionated bacterial respiration rates (for both free-living and particle-attached bacteria) in the hypoxic waters of station Y1 to the revised manuscript along with the bulk bacteria taxonomy data, although we do not have measurements for free-living bacteria abundance and taxonomy.

7. A similar argument for the lack of other organic compounds in the study.

Response: The reviewer is right about that there are other organic compounds that may also likely affect bacterial respiration. In the revised manuscript, we have compared PUAs with other organic compounds that would potentially affect bacterial activities in the hypoxia, such as 2-n-pentyl-4-quinolinol (PQ) and acylated homoserine lactones (AHLs). " A perennial bloom of PUA-producing diatoms in the PRE mouth (Wu and Li, 2016) should indicate the importance of PUAs for microbial activity here compared to many other organic compounds, such as 2-n-pentyl-4-quinolinol (Long et al., 2003) and acylated homoserine lactones (Hmelo et al., 2011).".

8. The conclusions must therefore be made more cautious, specific and these shortcomings commented on. Some speculative statements in the discussion and conclusion section need to be removed or rephrased.

Response: We agree with the reviewer on this. In the revised manuscript, we have rewritten the discussion and the conclusion sections.

9. There are parts of the method descriptions that need to be clarified, better specified or added. A major revision in this spirit is required to motivate publication.

Response: We have carefully addressed all these parts of descriptions concerning the methodology based on the reviewer's comments.

Detailed comments

10. r. 13-14 There is an extensive literature on eutrophication driven hypoxia in e.g. the Baltic Sea since 4 decades (cf. Cloern 2001). Please rephrase sentence accordingly

Response: Agree. In the revised manuscript, the sentence has been rewritten as " Eutrophication-driven coastal hypoxia is of great interest for decades".

11. r. 15. Do you mean ": : :water mainly dominated: : :".

Response: Agree.

12. r. 21 Please change "activity" to "..e.g. bacterial respiration and growth: : :) and revise the sentence.

Response: Done.

13. r. 35 Change to "..deoxygenation is also tightly: : :"

Response: Done.

14. r. 43-45 In most aquatic environments free living bacteria are dominating in numbers as well as biomass (e.g. Kirchman 2008). The reference is not convincingly showing that particle-associated bacterial dominate in terms of abundance or how the growth of particle associated bacteria was measured. Please revise the message. This also question the focus on particle associated bacteria in the manuscript.

Response: We agree with the reviewer that free-living bacteria are most dominant in many parts of the ocean. Meanwhile, our focus is on the high turbid coastal transition zone where particle-attached bacteria can be relatively more important. We have added a new reference of Lee et al (2015) to this sentence to show the dominance of PAB in some coastal regions. Lee, S., Lee, C., Bong, C., Narayanan, K., Sim, E.:

The dynamics of attached and free-living bacterial population in tropical coastal waters, Mar. Freshwater Res., 66, 701-710, 2015.

15. r. 49-51 In many cases free-living bacteria dominate the respiration (Robinson and leB Williams 2005). Both types of bacteria is preferably studied. This question the general relevance of the study.

Response: The reviewer is right about that free-living bacteria are important for community respiration in the open ocean. Meanwhile, what we focused on is the coastal transition zone, where particle-attached bacteria could be more important. Our field data indeed suggested that FLB respiration accounts for only 25-30% of the bulk bacterial community respiration in the hypoxic waters at station Y1. Nevertheless, we have rewritten these sentences to emphasize more on the general relevance of our study for both types of bacteria.

16. r. 64-65. However, many other organic compounds may drive the bacterial respiration. Please provide some reference showing to that extent PUF is contributing to bacterial respiration.

Response: We agree with the reviewer on this point. References for PUA contribution to bacterial metabolisms have been provided in the revised manuscript. "The strong effect of PUAs on bacterial growth, production, and respiration has been well demonstrated in the laboratory studies (Ribalet et al., 2008) and the field studies (Balestra et al., 2011; Edwards et al., 2015)."

17. r.68-76 I would prefer more explicit research questions to be addressed in this paragraph for clarity and coupling to performed experiments.

Response: Ok. We have rewritten this paragraph.

18. r. 84-85 Please define here what depths that were used for middle and bottom water categories (e.g. figure 6).

Response: The middle layer was at 12 m with the bottom layer at 25 m (4m above the

seafloor) for station Y1 (Figure 6). We have clarified this in the revised manuscript.

19. r. 87-88 Please define the abbreviations pPUA and dPUA.

Response: Agree. The pPUAs and dPUAs have been defined in the revised manuscript.

20. r. 97 Filtration and freezing of nutrient samples may release nutrients from broken cells.

Response: The influences of filtration and freezing/thaw on nutrient concentration should be negligible due to high nutrient concentration in the coastal system. They will only affect the oligotrophic open ocean waters with nanomolar nutrients (Li QP and Hansell DA, Anal Chim Acta, 611, 68-72, 2008).

21. r.124-129 How is the centrifugation and resuspension of particles influencing their morphology, attachments of PUAs and PABs?

Response: We believe that particle morphology and the attachments of PUAs and PABs will not be influenced by low-speed centrifugation (3000 rpm for one minute) or by a gently shaking for resuspension. The same approaches have been used to study particle-attached bacteria and particle-related compounds on sinking particles (Hmelo et al., 2011)

22. r. 131-132 Pleas provide the recovery efficiency of particle attached PUAs after the preparation procedure described.

Response: The recovery for the particle-adsorbed PUA should be 100% as the supernatants after centrifugations have all been added back to the final 50 ml centrifuge tube. We have clarified this in the revised manuscript.

23. r. 136-137. Please provide a reference where the method is validated.

Response: Done. References have been added to the revised manuscript. The protocol is modified from those of Edwards et al. (2015) and Wu and Li (2016).

24. r. 146-147 Freeze thawing may relate PUAs from living cells also.

Response: The reviewer is right that free-thawing would release PUAs from living cells. What we actually mean in the text is that we use the same determination method for undisrupted and disrupted PUAs although they are pre-treated differently (one with direct extraction method and the other with the freeze-thaw method). Anyway, we have clarified this in the revised manuscript.

25. r. 152 Should it be nmol per some volume or particle unit?

Response: Agree. It is the concentration of the undisturbed PUAs in the 50 mL sampling tube. We have clarified this in the revised manuscript.

26. r.160 Please specify what is meant by clean. What was the washing procedure?

Response: It means sterile. We have rewritten this in the revised manuscript. Generally, these bottles have been soaked in 10% HCl for 24 h, rinsed with deionized water for several times, and sterilized before use.

27. r.162 As presented here there was no true replication of the treatments?

Response: We actually have two replicates for each treatment. The sample in the 1-L Nalgene bottle had been transferred to two 0.5 L bottles for each treatment. We have clarified this in the revised manuscript.

28. r. 167 How does methanol included in the procedure affect bacterial abundance and activity? Any control or test for this? Please comment on relevance for natural conditions.

Response: The methanol has been added as a cosolvent for PUA (Franze et al., 2018). The methanol concentration of 0.05% in our experiment should not have a large effect on bacteria, since the previous study suggested that bacterial strains would not be significantly affected by methanol at a level below 1% (Patterson and Ricke, 2015). Patterson J.A., and S.C. Ricke, S.C., (2015) Effect of ethanol and methanol on growth

of ruminal bacteria Selenomonas ruminantium and Butyrivibrio fibrisolvens, Journal of Environmental Science and Health, Part B, 50:1, 62-67.

29. r. 176-179 Give some information on how close to natural conditions these final concentration of PUAs are.

Response: This PUA level was close to the hotspot PUAs concentration of 240 ïA▪mol L−1 found in a station near the PRE and was also comparable to the hotspot concentration of ∼60 ïA▪mol L−1 found in the western and subarctic North Atlantic (Edwards et al., 2015). We should emphasize that the concentration of PUA in the water-column is inhomogeneous due to the presence of particles. The hotspot concentration of PUA associated with these particles should be the PUA concentration in the volume of the water parcel taken up by the aggregation particles.

30. r.180 it is not obvious that turbidity will be detected if cells remain below about 109 cells cm3. Were the cell concentration measured by direct microscopy?

Response: We did not measure cell concentration during the experiments. The experiment is designed to only qualitatively assess the PAB response to different types of carbon sources. The culture duration of over 30 days should be long enough for significant bacterial growth (say with cell concentration well exceed the detection limit) to show up if the organic substrate could be used as a carbon source (Dong et al., 2015).

31. r. 181 Please provide a description on incubation conditions and length.

Response: Done. These experiments were performed in dark at room temperature for over 30 days. We have clarified this in the revised manuscript.

32. r. 185-196 The description of methodology is unclear. Please make clear if and how free-living bacterial abundance was measured? How is the methanol treatment accounting for bacteria from breaking particles? In addition, please provide a reference validating this method. Give some measure of the precision of the flow cytometer analysis and a relevant reference.

Response: We did not measure the free-living bacteria abundance. We only measure the abundance of the bulk water bacteria (>0.2 ïA■m) that includes both FLB and PAB. The method for bulk-water bacterial abundance has been added to the revised manuscript. We have also provided a reference for the flow cytometry method (Marie et al., 1997) as well as the relevant precision (CV%). The original text about methanol treatment was not well written, we have rewritten the sentence as "To break up particles and attached bacteria, 0.2 mL pure methanol was added to the 2 mL sample and vortexed".

Marie, D., Partensky, F., Jacquet, S. and Vaulot, D.: Enumeration and cell cycle analysis of natural populations of marine picoplankton by flow cytometry using the nucleic acid stain SYBR Green I, Appl. Environ. Microbiol. 63, 186-193, 1997

33. r.198-202 As no relevant pre-filtration is used you may include other organisms than bacteria in the respiration estimate. Please clarify and rephrase as needed.

Response: The reviewer is right about this. We did not perform any pre-filtration at this step. Besides the PAB, the particle aggregates of >25 $\mu$m would likely be consist of some phytoplankton and microplankton. So, the BR could be somewhat overestimated in our experiment. Nevertheless, this effect could be relatively small, given that bacterial respiration has been generally considered as the major contribution for community respiration. In addition, our particle samples were collected from the subsurface layer, which had negligible phytoplankters (as indicated by very low chlorophyll-a) and thus less zooplankton compared to the surface layer.

34. r. 208-209 What was the final TCA concentration in the sample. This does not follow the common procedure. Neither use of ethanol.

Response: Done. The final TCA concentration is 5%. The procedure of TCA step, as well as the use of ethanol, is based on the previous publication (Huang et al, 2018, doi: 10.1016/j.scitotenv.2018.03.222). We have clarified this in the revised manuscript.

35. r. 241 However, the lack of true replicates for the treatment (i.e. replicate 1-L Nalgene bottles per treatment) question a reliable result from the t-test.

Response: As we have responded to the previous point of this reviewer on No. 27, we have two replicates for each treatment. The sample in the 1-L Nalgene bottle had been transferred to two 0.5 L bottles for each treatment.

36. r. 251, Figure1. Consider to reverse the colour palette. More logical to have blue for well oxygenated and red for hypoxia. Change "constant" to "similar".

Response: Done. The figure has been revised as suggested by the reviewer. The word "constant" has been replaced by "similar" as well.

37. r. 261-263. Pleas provide a statistical test for the claimed difference and confidence intervals (error bars) in figure 4.

Response: Agree. In the revised manuscript, we have added statistical information to the difference for BR and BP. Error bars have also been provided in the revised figure.

38. r.265. Proved statistical results for the claimed difference between bacterial phyla. Do the same for other differences claimed throughout the manuscript.

Response: It is a typo. A statistical test is not doable for comparing different bacterial compositions. In the revised manuscript, we have corrected the sentence as "... was substantially different from those of X2 and X3". In addition, statistical information has been checked for each comparison throughout the manuscript.

39. r. 273 Should it be Bacteroidetes also here?

Response: Yes. We have revised as 4% of Bacteroidetes.

39-2. Figure 5. Please present the type of error bars used. Same for all figures with error bars.

Response: Error bars are the standard deviations. We have clarified these in the figure

legends of the revised manuscript.

40. r. 320-329 What can be considered significant differences as opposed to random variation in this analysis. Please motivate convincingly.

Response: Statistical information (t-value, n, and p-value) for comparing ïAğ-pro percentage between control and treatments have been added to the revised manuscript. However, a statistical test is not doable for comparing the difference of the bacterial community compositions. We have replaced the word "significant" with "substantial" in the revised manuscript.

41. r. 335 One month is an extremely long incubation. How relevant is this for the application to the natural environment?

Response: Bacterial utilization of organic carbons may depend on the nature of the organic compound. Bacteria may need a longer period to utilize refractory organic matters (ALK and PAH). On the other hand, our experiment goal is to qualitatively assess the possibility of PUA as a carbon source for PAB growth. The color change can be more easily appreciated after one-month for both PAH and ALK and thus allow us to compare the bacterial responses to different organic compounds (PUA, PAH, and ALK). There was no bacterial growth in the PUA medium throughout the one month should provide strong evidence that PUA was not used as a carbon source.

42. r. 360 Would be more informative to use a unit per particle or mass of particles? Litre of particles is unclear.

Response: The concentration of PUA in the water-column is inhomogeneous due to the presence of particles. The hotspot concentration of PUA should be the PUA concentration in the volume of the water parcel taken up by the aggregation particles. Therefore, particle volume is more informative and allows a better comparison of the hotspot concentration with the bulk water concentration.

43. r. 365 This assumes that PUA is a major substrate among all other organic compounds. Please provide some references on this matter and discuss it critically.

Response: We should emphasize that our focus here is to explore its role as a signal substance for PAB metabolism rather than as an organic carbon substrate for PAB growth. Actually, PUA accounts for only a small part of the particulate organic carbon (1-16%, Edwards et al., 2015). The specific arrangement of two double bonds and carbonyl chain makes PUA not a group of labile organic carbon for bacterial utilization. Anyway, we have rewritten the sentence to clarify this in the revised manuscript.

44. r. 368 This should be compared with the biomass of free living bacteria. They may also be elevated in the hypoxic water. I find the lack of measurement of free-living bacteria a short coming in the context of claiming importance of PABs.

Response: The reviewer is right about that free-living bacteria (FLB) may also be elevated in the hypoxic water. Our field data suggested that FLB respiration accounts for only 25-30% of the bulk bacterial community respiration in the hypoxic waters. We have added the bacterial respiration data of FLB (Figure S1) and the data of the community composition of bulk bacteria (Figure S2) to the revised manuscript.

45. r. 372-373. How is respiration by particle-attached bacteria distinguished from protozoa, phytoplankton and larger zooplankton? This is typically difficult to achieve. Comment in a critical manner.

Response: We cannot distinguish BR from respiration by phytoplankton and micro-zooplankton (large-zooplankton has been picked off already). Nevertheless, this effect could be relatively small given that bacterial respiration has been generally considered as the major contributor for community respiration (Robinson and Williams, 2005). In addition, our particle samples were collected from the subsurface layer, which had negligible phytoplankters (as indicated by very low chlorophyll-a) and thus less zooplankton compared to the surface layer.

46. r. 376-390. Given the apparent lack of true replication of the treatments (i.e.

replicate 1 L Nalgene bottles) the conclusions regarding treatment effects is highly uncertain. This needs a discussion.

Response: As we have responded to the previous point of this reviewer on No. 27, we have two replicates for each treatment. We have clarified this in the revised manuscript.

47. r. 389-390 Relevance in the natural environment assumes that the applied concentrations are relevant for those occurring in the natural environment. Please consider, discuss and modify the conclusion accordingly.

Response: We should emphasize that the concentration of PUA in the water-column is inhomogeneous due to the presence of particles. The micromolar level of PUA for incubation was chosen to represent the actual hotspot concentration of PUA (the PUA concentration in the volume of the water parcel taken up by the aggregation particles) not the mean PUA concentration (nanomolar level) in the bulk water. Anyway, we have clarified this and discussed them properly in the revised manuscript.

48. r. 391-392 I find it valuable to know if PUA stimulates bacterial activity whether as an organic substrate of metabolic signal substance. Please explain why only the latter would be ecologically important.

Response: We should note that PUA accounts only for a small percentage of the organic carbon (<16%, Edwards et al., 2015). Also, PUA can be toxic to some bacteria precluding its use as a carbon source. In addition, the specific arrangement of two double bonds and carbonyl chain make PUA not a group of labile organic carbon for bacterial utilization. Therefore, it has less ecological importance as a carbon substrate.

49. r. 392-394 Please use the same concentration unit for comparability of levels. Heptadienal alone used for the test may not be comparable to a mixture of different PUAs (i.e. concentration more than twice used in the combined concentration). Why was not the same mixture used for this experiment? Other methods like using labelled PUA and analyse for metabolism of those would better test the mechanism of PUA

effect.

Response: Agree. We have changed the unit to 200 ïĄ■mol/L in the revised manuscript. One reason for using heptadienal alone (C7) in the experiment is its lower toxicity compared to the other two (C8 and C10). Thus, C7 may be more likely used by bacteria if it can serve as a carbon source. Also, C7 is generally the dominant PUA over the large area of our study regions (Wu and Li 2016). Therefore, we focus on C7 alone rather than the mixture of various PUAs to qualitatively assess the bioavailability of PUAs to bacteria.

50. r. 405-407 Please consider that a few species within the Alteromonas phylum may be responsible for the observed response. PUA metabolism might not be a function attributed to the whole phyla. Rephrase the discussion accordingly.

Response: We agree with the reviewer that various bacterial species within the genus Altermonas may respond differently to the PUA treatments. We have revised the sentence as " . . . Our result was well consistent with the previous finding of the significant promotion effect of 13 or 106 ïĄ■molL-1 PUAs on Alteromonas hispanica from the pure culture experiment (Ribalet et al., 2008). An increase of PUAs would thus confer some of the ïĄğ–Pro (mainly special species within the genus Alteromonas, such as A. hispanica, Figure S2) a competitive advantage over other bacteria . . ."

51. r. 411-412 Please refer to what figure and test that show a difference between particle-attached and bulk bacteria.

Response: Ok. A figure has been provided in the supplement material to compare the community difference between particle-attached bacteria and the bulk bacteria (Figure S2A).

52. r. 427 How strongly are PUAs adsorbed to particles (i.e. chemical bonding)? How may this influence their potential to act as signalling molecules?

Response: It is still not well known about the mechanisms for PUA adsorption on particles. PUA may form a robust microzone around the particle, which would persist in the boundary layer and remain stable for some time (Juttner 2005). Juttner, F. (2005) Evidence that Polyunsaturated Aldehydes of Diatoms are Repellents for Pelagic Crustacean Grazers, Aquatic Ecology. 39, 271-282.

53. r. 439-441 I have not seen any analyses of the lipoxygenase hypothesis in the study? It is thus speculative and should be removed. Focus on conclusion that can be derived from the performed study.

Response: Agree. The related sentences have been removed in the revised manuscript.

54. r. 442-445 As there was no true replicates this conclusion should be made more cautious.

Response: As we have mentioned in our response to the previous point of this reviewer in No. 27, we did have two replicates for each treatment although more replicates are limited by the labor intensity of the experiment.

55. r.455-460 If this section should remain it need to be moved to the discussion section.

Response: Agree. The related sentences have been moved to the last part of the discussion section in the revised manuscript.

56. r. 460-464 The sudden appearance of PUFA is not connected to the previous sentence?. Again, this part is highly speculative and not part of conclusion from the study. Remove or move parts to the discussion.

Response: Agree. The sentences have been moved to the discussion section in the revised manuscript. To avoid disconnection between them, we have revised the sentences as "Eutrophication results in intense algae bloom with phytoplankton carbon sedimentation and accumulation in the coastal sediment, including PUFA compounds derived from the lipid production. Oxidation of these PUFA-rich organic particles during

. . ."

57. Literature cited Cloern, J. E. 2001. Our evolving conceptual model of the coastal eutrophication problem. Mar. Ecol.-Prog. Ser. 210: 223-253, doi.

Robinson, C., and P. J. Le B Williams. 2005. Respiration and its measurement in surface marine waters, p. 147-180. In P. A. del Giorgio and P. J. Williams, le B [eds.], Respiration in aquatic ecosystems. Oxford University Press.

Response: Agree. The mentioned references have been cited in the revised manuscript.

Please also note the supplement to this comment:
https://bg.copernicus.org/preprints/bg-2020-243/bg-2020-243-AC1-supplement.zip

---

## Author Comment (AC2) · 22 Oct 2020

1. However, there appeared to be some problems about the experimental design of this study and the manuscript fails to provide convincing results.

Response: We have carefully addressed the reviewer's comments on our experimental design and the related data and results. Please refer to our detailed responses to each of these specific comments below.

2. More details about the motivation and experiment procedure should be included and

clarified and the conclusions should be carefully justified.

Response: We thank the reviewer for this suggestion. We have carefully rewritten these parts of the manuscript in the introduction and the method sections to clarify the motivation of our study and provide the details of the experiment setup, operational procedure, and relevant methodology. We have also rewritten the conclusion section as the review suggested. Please refer to the specific comments of this reviewer below for details of our revisions in each section.

3. My major concerns about this manuscript is that the authors consider PAB as bacteria attached to particles with a size >25 $\mu$m in the microcosm incubation. A great variety of bacteria would be lost, which would affect the major conclusion of the manuscript. The abundance of bacteria attached to >25 $\mu$m particles would be significantly lower than that of free-living bacteria.

Response: We completely agree with the reviewer that a complete PAB community should be acquired using a smaller filtration such as 1 ïA▪m. Actually, in the high turbid estuarine waters of the PRE, PAB on the particle size of > 25 ïA▪m could be only about 20 percent of the PAB on the particle size of > 2 ïA▪m (Ge et al., 2020). Meanwhile, we should emphasize that our primary goal is to explore the mechanism for PUA affecting PAB variation and associated oxygen consumption in high turbidity and low oxygen regions of the PRE. Although the PUA was nanomolar in the bulk water, it can reach a micromolar level on the surface of the particles where they are produced. The hotspot PUA concentration accumulated on the particle surface may increase along with the growth of particle aggregations, varying from 1 ïA▪mol L-1 to more than 100 ïA▪mol L-1 (Edwards et al., 2015). Therefore, we chose the larger aggregates with the particle size of > 25 ïA▪m in the hypoxic waters to perform the PUA-amended experiments, in order to better explore the PUA effects on PAB in the hypoxic waters. We agree that a more systematic study in the future may need to investigate PUA impacts on PAB associated with the particle size of >1 ïA▪m. Future study may also need to explore the impact of nanomolar PUAs on the free-living bacteria in the background bulk waters.

4. Also, is this particle size proper for measurement of polyunsaturated aldehydes?

Response: It is specific for particle-adsorbed PUAs on particles of > 25 ïA■m. A previous study by Edwards et al (2015) uses an even larger size of 50 ïA■m for collecting sinking particles for PAB and the associated estimation of hotspot PUAs concentration.

5. In addition, it is not clear why the authors choose 1 or 100 $\mu$mol L-1 but not the background value for the incubation.

Response: We should emphasize that the concentration of PUA in the water-column is inhomogeneous due to the presence of particles. Although the PUA was nanomolar in the bulk water, it can reach a micromolar level on the surface of the particles where they are produced. The micromolar level of PUA for incubation was chosen to represent the actual hotspot concentration of PUA (the PUA concentration in the volume of the water parcel taken up by the aggregation particles) not the mean PUA concentration (nanomolar level) in the bulk water.

6. Moreover, bacterial community of the initial inoculates was lacking.

Response: In the revised manuscript, we have provided the initial bacterial community data (T=0) for the experiment in the supplementary material (Figure S2).

7. Personally, the most interesting part of this study is the role of polyunsaturated aldehydes-enhanced bacterial oxygen demand for the seasonal hypoxia. Thus, it is important to know to what extent different concentration of polyunsaturated aldehydes affect bacterial growth and respiration. Although the authors provide discussion on this, more analyses including the selection of background concentration of polyunsaturated aldehydes and testing on pure isolates are needed.

Response: We thank the reviewer for this suggestion. In the revised manuscript, we have provided a discussion on the impact of the background nanomolar level of PUAs on bacteria activity. "The effect of background nanomolar PUAs on free-living bacteria was not explored during our study. Previous studies of the coastal bacterial communities in the NW Mediterranean Sea suggested that 7.5 nmolL-1 PUAs would have a different effect on the metabolic activity of distinct bacterial groups although bulk bacterial abundance remained unchanged (Balestra et al., 2011). In particular, the metabolic activity of ïAğ-Pro was least affected by nanomolar PUAs, although those of Bacteroidetes and Rhodobacteraceae were markedly depressed (Balestra et al., 2011). Meanwhile, the daily addition of 1 nmolL-1 PUAs was found to not affect bacterial abundance and community composition during a mesocosm experiment in the Bothnian Sea (Paul et al., 2012)."

8. It is unclear why the authors studied the effect of polyunsaturated aldehydes of bacterial communities. The importance of polyunsaturated aldehydes was not properly and clearly presented. For examples, although the authors mentioned the effect of polyunsaturated aldehyde on marine microorganisms, detailed processes and mechanisms are not provided.

Response: We agree with the reviewer on this. In the revised manuscript, we have carefully rewritten this part by emphasizing the importance of PUAs on the microbial community and the associated mechanisms. " Phytoplankton-derived polyunsaturated aldehydes (PUAs) are known to affect marine microorganisms over various trophic levels by acting as infochemicals and/or chemical defenses (Ribalet et al., 2008; Ianora and Miralto, 2010; Edwards et al., 2015; Franzè et al., 2018). The strong effect of PUAs on bacterial growth, production, and respiration has been well demonstrated in laboratory studies (Ribalet et al., 2008) and in the field studies (Balestra et al., 2011; Edwards et al., 2015). A perennial bloom of PUA-producing diatoms in the PRE mouth (Wu and Li, 2016) should indicate the importance of PUAs for microbial activity here compared to many other organic compounds, such as 2-n-pentyl-4-quinolinol (Long et al., 2003) and acylated homoserine lactones (Hmelo et al., 2011). A nanomolar level of PUAs recently reported in the coastal waters outside the PRE was hypothesized to affect oxygen depletion by promoting microbial utilization of organic matters in the bottom waters (Wu and Li, 2016), while the actual role of PUAs on bacterial metabolism

within the bottom hypoxia remains largely unexplored.".

9. Line 43: Please provide reference for the higher abundance of PAB compared to FLB.

Response: Done. The revised sentence is written as " In some coastal waters, PAB could be more abundant than the FLB with higher metabolic activity and may affect coastal carbon cycle through organic matter remineralization (Garneau et al., 2009; Lee et al., 2015)."

10. Line 59-60: How polyunsaturated aldehyde affects marine microorganisms? Please provide more details.

Response: Done. We have rewritten the sentence as " Phytoplankton-derived polyun-saturated aldehydes (PUAs) are known to affect marine microorganisms over various trophic levels by acting as infochemicals and/or chemical defenses (Ribalet et al., 2008; Ianora and Miralto, 2010; Edwards et al., 2015; Franzè et al., 2018). ".

11. Line 63: What is the meaning of affect oxygen depletion? Is this a promoting or inhibiting process?

Response: Done. The sentence has been rewritten as "... affect oxygen depletion by promoting microbial utilization of organic matters ...".

12. Line 72-73: It is strange to place this sentence here. Why PUAs did not serve as carbon source?

Response: Agree. We have moved the sentence to the method and result sections. Firstly, PUA can be toxic to some bacteria precluding its use as a carbon source. Secondly, the specific arrangement of two double bonds and carbonyl chain make PUA not a group of labile organic carbon for bacterial utilization. There were other studies supporting that PUA could not serve as a carbon source for bacterial growth (Ribalet., 2008; Edwards et al., 2015).

13. Line 81: Is that "July 2nd"?

Response: It is July 2nd. We have corrected this typo in the revised manuscript.

14. Line 88: what do you mean by "pPUAs and dPUAs"?

Response: Done. The abbreviations of pPUAs and dPUAs have been defined in the revised manuscript.

15. Line 265: I did not see description of methods about the bacterial community analysis in bottom waters of X1, X2, X3 and PAB on particles of >25 $\mu$m.

Response: Done. We have added the method descriptions of these data to the section of 2.7.4 in the revised manuscript. " DNA samples for the bulk bacteria (>0.2 $\mu$m) and PAB on particles of > 25 $\mu$m at station Y1 were also collected for bacterial community analysis using the same method described above. Methods for the bulk water bacterial community analyses at station X1, X2, and X3 during the 2016 cruise can be found in the published paper of Xu et al. (2018).".

16. Line 276: Please provide the concentration of pPUAs and dPUAs.

Response: Done. The mean concentration of pPUAs and dPUAs has been provided in the revised manuscript.

17. Line 278: shown Line 391: According to the results, low dose (1 $\mu$mol) of PUAs can stimulate the growth of PAB, significantly different from that of high dose (100 $\mu$mol) treatment. However, the test of PUAs as organic carbon source was conducted with 200 $\mu$mol of PUAs. I guess such a high concentration would adversely affect bacteria growth, while the low dose PUAs is likely to be used as organic sources.

Response: The 200 $\mu$M PUAs used in the test of carbon source possibility was to assure the same level of organic carbon substrate as those for ALK and PAHs. Bacteria may need a longer time and a higher substrate concentration to utilize these refractory organic matters (ALK and PAHs). Although we have no test for the low-dose PUAs, the

previous study has suggested that low-level PUAs (1 $\mu$M and 10 $\mu$M) were not used as a carbon source by bacteria (Edwards et al., 2015).

18. Line 686: data of panel D are reproduced from Ribalet et al., 2008. Is this panel E? No methods were provided for growth of Alteromonas hispanica MOLA151.

Response: Agree. It should be panel E. We have corrected this in the revised manuscript. We have also provided the growth method of A. hispanica MOLA151.

19. Line 448: Since bacteria on >25 $\mu$m particles can be low, hypothesis on signaling molecules may be tuned down.

Response: Agree. We have rewritten this sentence as "... we hypothesize that PUAs may likely act as signaling molecules for coordination among the high-density PAB below the salt-wedge, which will potentially allow bacteria such as Alteromonas to thrive in degrading particulate organic matters ... ".

Please also note the supplement to this comment:
https://bg.copernicus.org/preprints/bg-2020-243/bg-2020-243-AC2-supplement.pdf

―――――――――――――――――――――――

**Supplement:**

[revised manuscript text omitted]

FLB could only take up less than 25-30 % of the bulk bacterial community respiration in the hypoxic

waters. Therefore, it is important to address the linkage between the high-density PAB and the high level of

particle-adsorbed PUAs associated with the suspended particles in the low-oxygen waters.

Interestingly, our PUA-amended experiments for PAB retrieved from the low-oxygen waters revealed

distinct responses of PAB to different doses of PUAs treatments with an increase in cell growth in response

to low-dose PUAs (1 μmol L$^{-1}$) but an elevated cell-specific metabolic activity including bacterial

respiration and production in response to high-dose PUAs (100 μmol L$^{-1}$). An increase in cell density of

PAB by low-dose PUAs could likely reflect the stimulating effect of PUAs on PAB growth. This finding

was consistent with the previous report of a PUAs level of 0-10 μmol L$^{-1}$ stimulating respiration and cell

growth of PAB in sinking particles of the open ocean (Edwards et al., 2015). The negligible effect of

low-dose PUAs on bacterial community structure in our experiments was also in good agreement with

those found for PAB from sinking particles (Edwards et al., 2015). However, we do not see the inhibitory

effect of 100 μmol L$^{-1}$ PUAs on PAB respiration and production previously found in the open ocean

(Edward et al., 2015). Instead, the stimulating effect for high-dose PUAs on bacterial respiration and

production was even stronger with ~50% of increments. The bioactivity of PUAs on bacterial strains could

likely arise from its specific arrangement of two double bonds and carbonyl chain (Ribalet et al., 2008).

Our findings strongly support the important role of PUAs in enhancing bacterial oxygen utilization in the

low-oxygen waters.

The effect of background nanomolar PUAs on free-living bacteria was not explored during our study.

Previous studies of the coastal bacterial communities in the NW Mediterranean Sea suggested that 7.5

[revised manuscript text omitted]

779                                          **Figure 1**

780

[Figure]

[Figure]

**Figure 2**

[Figure]

784

785

786                                                           **Figure 3**

[Figure]

787

788

789

790

**Figure 4**

791

[Figure]

[Figure]

**Figure 6**

[Figure]

**Figure 7**

[Figure]

804

805

806                                    **Figure 8**

[Figure]

**Figure 9**

---

## Author Response (AR1)

November 2nd, 2020

Dear Editor,

Attached is a revised version of our manuscript, "Impacts of biogenic polyunsaturated aldehydes on metabolism and community composition of particle-attached bacteria in coastal hypoxia" by Zhengchao Wu et al.

We greatly appreciate the comments and suggestions provided by the reviewers and editor. They have been very constructive, contributing significantly to improve the overall quality of our paper. We have carefully addressed all their points in the revised manuscript, and have detailed our changes in the response to reviewers (below). We hope that you will now find our manuscript suitable for publication.

Sincerely yours,

Qian Li South China Sea Institute of Oceanology Chinese Academy of Sciences, Guangzhou, China Phone: 011-86-20-84454476 Email: qianli@scsio.ac.cn

**Response to Anonymous Referee #1**

**General comments**

1. The manuscript is generally well written with proper language although some sentences need improvements. Results are well presented in the figures, but some complement of error bars are needed.

Response: Thanks to the reviewer for constructive comments. The manuscript has been proofread by a native English speaker to correct grammar errors and to improve the written language. We have redone these figures by including the error bars in the revised manuscript.

2. I lack a convincing motivation to the importance of PUAs compared to the multitude of other organic compounds, and the importance of particle-attached bacteria as compared to free-living. No direct comparison with other organic compounds or free-living bacteria is done in the study.

Response: The reviewer is right that there are other organic compounds that may also likely affect bacterial respiration, such as 2-n-pentyl-4-quinolinol (Long et al., 2003) and acylated homoserine lactones (Hmelo et al., 2011). However, a perennial bloom of PUA-producing diatoms in the PRE mouth (Wu and Li, 2016) should argue for the importance of PUAs for microbial activity here compared to many other organic compounds. The reviewer is also right that free-living bacteria are important for community respiration in the ocean. However, our focus here is on coastal zones. Our field measurements suggested that bacterial respiration in the hypoxic waters was largely contributed by particle-attached bacteria (>0.8  $\mu$ m) with FLB (0.2-0.8  $\mu$ m) only accounting for 25-30% of the total rates. We have provided the data of the size-fractionated bacterial respiration rates in Figure S1 of the revised manuscript.

3. A general importance of PUAs and particle attached bacteria should be tuned down in the discussion and conclusion. The effects of PUA on particle-attached bacteria is still of value as such.

**Response:** OK. We have rewritten the relevant sentences in the discussion and the conclusion sections as suggested by the reviewer.**

4. A short-coming of the experimental design is a lack of true replication of the treatments.

**Response:** We actually have two replicates for each treatment. The original text was not well written. We have clarified this in the revised manuscript.**

5. In addition, the fact that only one season has been investigated.

**Response:** The hypoxia could only occur during the summer in our study region. Therefore, a seasonality of hypoxia suggested by the reviewer may be unnecessary.

6. I also miss proper measurement of the abundance, acidity and taxonomy of free-living bacterial to put the claimed influence of particle attached bacteria in perspective.

Response: We have added data of the size-fractionated bacterial respiration rates (for both free-living and particle-attached bacteria) in the hypoxic waters of station Y1 to the revised manuscript along with the bulk bacteria taxonomy data, although we do not have measurements for free-living bacteria abundance and taxonomy.

7. A similar argument for the lack of other organic compounds in the study.

Response: The reviewer is right about that there are other organic compounds that may also likely affect bacterial respiration. In the revised manuscript, we have compared PUAs with other organic compounds that would potentially affect bacterial activities in the hypoxia, such as 2-n-pentyl-4-quinolinol (PQ) and acylated homoserine lactones (AHLs). "A perennial bloom of PUA-producing diatoms in the PRE mouth (Wu and Li, 2016) may indicate the importance of PUAs for microbial activity here compared to many other organic compounds, such as 2-n-pentyl-4-quinolinol (Long et al., 2003) and acylated homoserine lactones (Hmelo et al., 2011)."

8. The conclusions must therefore be made more cautious, specific and these shortcomings commented on. Some speculative statements in the discussion and conclusion section need to be removed or rephrased.

**Response: We agree with the reviewer on this. In the revised manuscript, we have carefully rewritten the discussion and the conclusion sections.**

9. There are parts of the method descriptions that need to be clarified, better specified or added. A major revision in this spirit is required to motivate publication.

**Response: We thank the review for these comments. We have carefully rewritten all the parts of descriptions of the relevant methodology suggested by the reviewer.**

**Detailed comments**

10. r. 13-14 There is an extensive literature on eutrophication driven hypoxia in e.g. the Baltic Sea since 4 decades (cf. Cloern 2001). Please rephrase sentence accordingly

**Response:** Agree. In the revised manuscript, the sentence has been rewritten as "Eutrophication-driven coastal hypoxia is of great interest for decades". 11. r. 15. Do you mean ": : : water mainly dominated: : :".

**Response: Agree.**

12. r. 21 Please change "activity" to "..e.g. bacterial respiration and growth: : :) and revise the sentence.

**Response: Done.**

13. r. 35 Change to ".. deoxygenation is also tightly: : :"

**Response: Done.**

14. r. 43-45 In most aquatic environments free living bacteria are dominating in numbers as well as biomass (e.g. Kirchman 2008). The reference is not convincingly showing that particle-associated bacterial dominate in terms of abundance or how the growth of particle associated bacteria was measured. Please revise the message. This also question the focus on particle associated bacteria in the manuscript.

Response: We agree with the reviewer that free-living bacteria are most dominant in many parts of the ocean. However, our focus is on the high turbid coastal transition zone where particle-attached bacteria can be relatively more important. We have added another reference of Lee et al (2015) to this sentence to show the importance of PAB in coastal regions. Lee, S., Lee, C., Bong, C., Narayanan, K., Sim, E.: The dynamics of attached and free-living bacterial population in tropical coastal waters, Mar. Freshwater Res., 66, 701-710, 2015.

15. r. 49-51 In many cases free-living bacteria dominate the respiration (Robinson and leB Williams 2005). Both types of bacteria is preferably studied. This question the general relevance of the study.

Response: The reviewer is right about that free-living bacteria are more important for community respiration in the open ocean. However, what we focused on is the coastal transition zone, where particle-attached bacteria could be important. Our field measurements suggested that bacterial respiration in the hypoxic waters was largely contributed by particle-attached bacteria (>0.8  $\mu$ m) with FLB (0.2-0.8  $\mu$ m) only accounting for 25-30% of the total rates. We have added these results to the revised manuscript. We have also rewritten these sentences to emphasize more on the general relevance of our study for both types of bacteria.

16. r. 64-65. However, many other organic compounds may drive the bacterial respiration. Please provide some reference showing to that extent PUF is contributing to bacterial respiration.

Response: We agree with the reviewer on this point. References for PUAs contribution to bacterial

metabolisms have been provided in the revised manuscript. "The strong effect of PUAs on bacterial growth, production, and respiration has been well demonstrated in the laboratory studies (Ribalet et al., 2008) and the field studies (Balestra et al., 2011; Edwards et al., 2015)."

17. r.68-76 I would prefer more explicit research questions to be addressed in this paragraph for clarity and coupling to performed experiments.

Response: Ok. We have rewritten this paragraph as "...There are three specific questions to address here: What are the relative roles of PAB and FLB on bacterial respiration in the hypoxic waters? What are the actual levels of PUAs in the hypoxic waters? What are the responses of PAB to PUAs in the hypoxic waters? For the first question, size-fractionated bacterial respiration rates were estimated for both FLB (0.2-0.8  $\mu$ m) and PAB (>0.8  $\mu$ m) in the hypoxic waters. For the second question, the concentrations of particulate and dissolved PUAs within the hypoxic waters were measured in the field. Besides, the hotspot PUAs concentration associated with the suspended particles within the hypoxic waters was directly quantified for the first time using large-volume filtration and subsequent on-site derivation and extraction. For the third question, field PUAs-amended incubation experiments were conducted for PAB (>25  $\mu$ m) retrieved from the low-oxygen waters. We focused on particles of >25  $\mu$ m to better explore the role of PUAs on PAB given the actual levels of PUAs hotspots, to assess the PAB responses (including bacterial abundance, respiration, production, and community composition) to the exogenous PUAs in the hypoxic waters..."

18. r. 84-85 Please define here what depths that were used for middle and bottom water categories (e.g. figure 6).

Response: The middle layer was at 12 m with the bottom layer at 25 m (4m above the seafloor) for station Y1 (Figure 6). We have clarified this in the revised manuscript.

19. r. 87-88 Please define the abbreviations pPUA and dPUA.

**Response: Agree. The pPUAs and dPUAs have been defined in the revised manuscript.**

20. r. 97 Filtration and freezing of nutrient samples may release nutrients from broken cells.

Response: The influences of filtration and freezing/thaw on nutrient concentration should be negligible due to high nutrient concentration in the coastal system. They will only affect the oligotrophic open ocean waters with nanomolar nutrients (Li QP and Hansell DA, Anal Chim Acta, 611, 68-72, 2008).

21. r.124-129 How is the centrifugation and resuspension of particles influencing their morphology, attachments of PUAs and PABs?

Response: We believe that particle morphology and the attachments of PUAs and PABs will not be influenced by low-speed centrifugation (3000 rpm for one minute) or by a gently shaking for resuspension. The same approaches have been used to study particle-attached bacteria and particle-related compounds on sinking particles (Hmelo et al., 2011)

22. r. 131-132 Pleas provide the recovery efficiency of particle attached PUAs after the preparation procedure described.

Response: The recovery for the particle-adsorbed PUAs should be 100% as the supernatants after centrifugations have all been added back to the final 50 ml centrifuge tube. We have clarified this in the revised manuscript.

23. r. 136-137. Please provide a reference where the method is validated.

Response: Done. References have been added to the revised manuscript. The protocol is modified from those of Edwards et al. (2015) and Wu and Li (2016).

24. r. 146-147 Freeze thawing may relate PUAs from living cells also.

Response: The reviewer is right that free-thawing would release PUAs from living cells. What we actually mean in the text is that we use the same determination method for undisrupted and disrupted PUAs although they are pre-treated differently (one with direct extraction method and the other with the freeze-thaw method). Anyway, we have clarified this in the revised manuscript.

25. r. 152 Should it be nmol per some volume or particle unit?

Response: Agree. It is the concentration of the undisturbed PUAs in the 50 mL sampling tube. We have clarified this in the revised manuscript.

26. r.160 Please specify what is meant by clean. What was the washing procedure?

Response: It means sterile. We have rewritten this in the revised manuscript. Generally, these bottles have been soaked in 10% HCl for 24 h, rinsed with deionized water for several times, and sterilized before use.

27. r.162 As presented here there was no true replication of the treatments?

Response: We actually have two replicates for each treatment. The sample in the 1-L Nalgene bottle had been transferred to two 0.5 L bottles for each treatment. We have clarified this in the revised manuscript.

28. r. 167 How does methanol included in the procedure affect bacterial abundance and activity? Any control or test for this? Please comment on relevance for natural conditions.

Response: The methanol has been added as a cosolvent for PUAs (Franze et al., 2018). The methanol concentration of 0.05% in our experiment should not have a large effect on bacteria, since the previous study suggested that bacterial strains would not be significantly affected by methanol at a level below 1% (Patterson and Ricke, 2015).

Patterson J.A., and S.C. Ricke, S.C., (2015) Effect of ethanol and methanol on growth of ruminal bacteria *Selenomonas ruminantium* and *Butyrivibrio fibrisolvens*, Journal of Environmental Science and Health, Part B, 50:1, 62-67.

29. r. 176-179 Give some information on how close to natural conditions these final concentration of PUAs are.

Response: The PUAs level was close to the hotspot PUAs concentration of 240  $\mu$ mol L-1 found in a station near the PRE and was also comparable to the hotspot concentration of ~26  $\mu$ mol L-1 found in the temperate west North Atlantic (Edwards et al., 2015). We should emphasize that the concentration of PUAs in the water-column is inhomogeneous due to the presence of particles. The hotspot concentration of PUAs associated with these particles should be the PUAs concentration in the volume of the water parcel displaced by the aggregation particles.

30. r.180 it is not obvious that turbidity will be detected if cells remain below about  $10^9$  cells cm3. Were the cell concentration measured by direct microscopy?

Response: We did not measure cell concentration during the experiments. The experiment is designed to only qualitatively assess the PAB response to different types of carbon sources. The culture duration of over 30 days should be long enough for significant bacterial growth (say with cell concentration well exceed the detection limit) to show up if the organic substrate could be used as a carbon source (Dong et al., 2015, doi:10.5194/bg-12-2163-2015).

31. r. 181 Please provide a description on incubation conditions and length.

**Response: Done. These experiments were performed in dark at room temperature for over 30 days.** We have clarified this in the revised manuscript.

32. r. 185-196 The description of methodology is unclear. Please make clear if and how free-living bacterial abundance was measured? How is the methanol treatment accounting for bacteria from breaking particles? In addition, please provide a reference validating this method. Give some measure of the precision of the flow cytometer analysis and a relevant reference.

Response: We did not measure the free-living bacteria abundance. We only measure the abundance of the bulk water bacteria (>0.2  $\mu$ m) that includes both FLB and PAB. The method for bulk-water bacterial abundance has been added to the revised manuscript. We have also provided a reference for the flow cytometry method (Marie et al., 1997) as well as the relevant precision (CV%). The original text about methanol treatment was not well written, we have rewritten the sentence as "To break up particles and attached bacteria, 0.2 mL pure methanol was added to the 2 mL sample and vortexed".

Marie, D., Partensky, F., Jacquet, S. and Vaulot, D.: Enumeration and cell cycle analysis of natural populations of marine picoplankton by flow cytometry using the nucleic acid stain SYBR Green I, Appl. Environ. Microbiol. 63, 186-193, 1997

33. r.198-202 As no relevant pre-filtration is used you may include other organisms than bacteria in the respiration estimate. Please clarify and rephrase as needed.

Response: The reviewer is right about this. We did not perform any pre-filtration at this step. Besides the PAB, the particle aggregates of >25  $\mu$ m would likely consist of some phytoplankton and microzooplankton. So, the BR could be overestimated in our experiment. However, this effect could be relatively small, given that the chlorophyll-a of the raw seawater (hypoxic waters in the subsurface layer) was very low and there was not much microzooplankton in the sample (confirmed by FlowCAM). We have clarified this in the revised manuscript.

34. r. 208-209 What was the final TCA concentration in the sample. This does not follow the common procedure. Neither use of ethanol.

**Response: Done. The final TCA concentration is 5%. The procedure of TCA step, as well as the use of ethanol, is based on the previous publication (Huang et al, 2018, doi: 10.1016/j.scitotenv.2018.03.222). We have clarified this in the revised manuscript.**

35. r. 241 However, the lack of true replicates for the treatment (i.e. replicate 1-L Nalgene bottles per treatment) question a reliable result from the *t*-test.

Response: As we have responded to the previous point of this reviewer on No. 27, we have two replicates for each treatment. The sample in the 1-L Nalgene bottle had been transferred to two 0.5 L bottles for each treatment.

36. r. 251, Figure 1. Consider to reverse the colour palette. More logical to have blue for well oxygenated and red for hypoxia. Change "constant" to "similar".

Response: Done. The figure has been revised as suggested by the reviewer. The word "constant" has been replaced by "similar" as well.

37. r. 261-263. Pleas provide a statistical test for the claimed difference and confidence intervals (error bars) in figure 4.

**Response: Agree. In the revised manuscript, we have added statistical information to the difference for BR and BP. Error bars have also been provided in the revised figure.**

38. r.265. Proved statistical results for the claimed difference between bacterial phyla. Do the same for other differences claimed throughout the manuscript.

Response: It is a typo. A statistical test is not doable for comparing different bacterial compositions. In the revised manuscript, we have corrected the sentence as "... was substantially different from those of X2 and X3". In addition, statistical information has been checked for each comparison throughout the manuscript.

39. r. 273 Should it be Bacteroidetes also here?

**Response: Yes. We have revised as 4% of Bacteroidetes.**

39-2. Figure 5. Please present the type of error bars used. Same for all figures with error bars.

**Response: Error bars are the standard deviations. We have clarified these in the figure legends of the revised manuscript.**

40. r. 320-329 What can be considered significant differences as opposed to random variation in this analysis. Please motivate convincingly.

Response: Statistical information (t-value, n, and p-value) for comparing  $\gamma$ -pro percentage between control and treatments have been added to the revised manuscript. However, a statistical test is not doable for comparing the difference of the bacterial community compositions. We have replaced the word "significant" with "substantial" in the revised manuscript.

41. r. 335 One month is an extremely long incubation. How relevant is this for the application to the natural environment?

Response: Bacterial utilization of organic carbons may depend on the nature of the organic compound. Bacteria may need a longer period to utilize refractory organic matters (ALK and PAH). On the other hand, our experiment goal is to qualitatively assess the possibility of PUA as a carbon source for PAB growth. The color change can be more easily appreciated after one-month for both PAH and ALK and thus allow us to compare the bacterial responses to different organic

compounds (PUA, PAH, and ALK). There was no bacterial growth in the PUA medium throughout the one month should provide strong evidence that PUA was not used as a carbon source.

42. r. 360 Would be more informative to use a unit per particle or mass of particles? Litre of particles is unclear.

Response: The concentration of PUAs in the water-column is inhomogeneous due to the presence of particles. The hotspot concentration of PUAs should be the PUAs concentration in the volume of the water parcel displaced by the aggregation particles. Therefore, particle volume is more informative and allows a better comparison of the hotspot concentration with the bulk water concentration.

43. r. 365 This assumes that PUA is a major substrate among all other organic compounds. Please provide some references on this matter and discuss it critically.

Response: We should emphasize that our focus here is to explore its role as a signal substance for PAB metabolism rather than as an organic carbon substrate for PAB growth. Actually, PUAs accounts for only a small part of the particulate organic carbon (1-16%, Edwards et al., 2015). The specific arrangement of two double bonds and carbonyl chain makes PUAs not a group of labile organic carbon for bacterial utilization. Anyway, we have rewritten the sentence to clarify this in the revised manuscript.

44. r. 368 This should be compared with the biomass of free living bacteria. They may also be elevated in the hypoxic water. I find the lack of measurement of free-living bacteria a short coming in the context of claiming importance of PABs.

Response: The reviewer is right about that free-living bacteria (FLB) may also be elevated in the hypoxic water. Our field data suggested that FLB respiration accounts for only 25-30% of the total bacterial community respiration in the hypoxic waters. In the revised manuscript, we have provided the data of the size-fractionated respiration rates for FLB and PAB (Figure S1), as well as the data of the community composition of bulk bacteria (Figure S2).

45. r. 372-373. How is respiration by particle-attached bacteria distinguished from protozoa, phytoplankton and larger zooplankton? This is typically difficult to achieve. Comment in a critical manner.

Response: We cannot distinguish BR from the respirations of phytoplankton and microzooplankton (larger zooplankton has been picked off already). However, this effect could be relatively small, since the raw sweater in the hypoxic zone had very low chlorophyll-a and there was virtually not much microzooplankton in the sample (confirmed by FlowCAM). We have

**clarified these in the revised manuscript.**

46. r. 376-390. Given the apparent lack of true replication of the treatments (i.e. replicate 1 L Nalgene bottles) the conclusions regarding treatment effects is highly uncertain. This needs a discussion.

**Response: As we have responded to the previous point of this reviewer on No. 27, we have two replicates for each treatment. We have clarified this in the revised manuscript.**

47. r. 389-390 Relevance in the natural environment assumes that the applied concentrations are relevant for those occurring in the natural environment. Please consider, discuss and modify the conclusion accordingly.

Response: We should emphasize that the concentration of PUAs in the water-column is inhomogeneous due to the presence of particles. The micromolar level of PUAs for incubation was chosen to represent the actual hotspot concentration of PUAs (the PUAs concentration in the volume of the water parcel displaced by the aggregation particles) not the mean PUAs concentration (nanomolar level) in the background seawater. Anyway, we have clarified this and discussed them properly in the revised manuscript.

48. r. 391-392 I find it valuable to know if PUA stimulates bacterial activity whether as an organic substrate of metabolic signal substance. Please explain why only the latter would be ecologically important.

Response: We should note that PUAs accounts only for a small percentage of the organic carbon (<16%, Edwards et al., 2015). Also, PUAs can be toxic to some bacteria precluding its use as a carbon source. In addition, the specific arrangement of two double bonds and carbonyl chain make PUAs not a group of labile organic carbon for bacterial utilization. Therefore, it has less ecological importance as a carbon substrate.

49. r. 392-394 Please use the same concentration unit for comparability of levels. Heptadienal alone used for the test may not be comparable to a mixture of different PUAs (i.e. concentration more than twice used in the combined concentration). Why was not the same mixture used for this experiment? Other methods like using labelled PUA and analyse for metabolism of those would better test the mechanism of PUA effect.

Response: Agree. We have changed the unit (200  $\mu$ mol L-1) in the revised manuscript. One reason for using heptadienal alone (C7) in the experiment is its lower toxicity compared to the other two (C8 and C10). Thus, C7 may be more likely used by bacteria if it can serve as a carbon source. Also, C7 is generally the dominant PUAs over the large area of our study regions (Wu and Li 2016). Therefore, we focus on C7 alone rather than the mixture of various PUAs to qualitatively assess the bioavailability of PUAs to bacteria. 50. r. 405-407 Please consider that a few species within the *Alteromonas phylum* may be responsible for the observed response. PUA metabolism might not be a function attributed to the whole phyla. Rephrase the discussion accordingly.

Response: We agree with the reviewer that various bacterial species within the genus *Altermonas* may respond differently to the PUAs treatments. We have revised the sentence as "...Our result was well consistent with the previous finding of the significant promotion effect of 13 or 106  $\mu$ mol L-1 PUAs on *Alteromonas hispanica* from the pure culture experiment (Ribalet et al., 2008). An increase of PUAs could thus confer some of the  $\gamma$ -Pro (mainly special species within the genus *Alteromonas*, such as *A. hispanica*, Figure S2B) a competitive advantage over other bacteria ..."

51. r. 411-412 Please refer to what figure and test that show a difference between particle-attached and bulk bacteria.

**Response:** Ok. A figure has been provided in the supplement material to compare the community difference between particle-attached bacteria and the bulk bacteria (Figure S2A).**

52. r. 427 How strongly are PUAs adsorbed to particles (i.e. chemical bonding)? How may this influence their potential to act as signalling molecules?

**Response: It is still not well known about the mechanisms for PUAs adsorption on particles. PUAs may form a robust microzone around the particle, which would persist in the boundary layer and remain stable for some time (Juttner 2005).**

Juttner, F. (2005) Evidence that Polyunsaturated Aldehydes of Diatoms are Repellents for Pelagic Crustacean Grazers, Aquatic Ecology. 39, 271-282.

53. r. 439-441 I have not seen any analyses of the lipoxygenase hypothesis in the study? It is thus speculative and should be removed. Focus on conclusion that can be derived from the performed study.

**Response: Agree. The related sentences have been deleted in the revised manuscript.**

54. r. 442-445 As there was no true replicates this conclusion should be made more cautious.

Response: As we have mentioned in our response to the previous point of this reviewer in No. 27, we did have two replicates for each treatment although more replicates are limited by the labor intensity of the experiment.

55. r.455-460 If this section should remain it need to be moved to the discussion section.

**Response: Agree. The related sentences have been moved to the last part of the discussion section**

**in the revised manuscript.**

56. r. 460-464 The sudden appearance of PUFA is not connected to the previous sentence?. Again, this part is highly speculative and not part of conclusion from the study. Remove or move parts to the discussion.

Response: Agree. The sentences have been moved to the discussion section in the revised manuscript. To avoid disconnection between them, we have revised the sentences as "...Eutrophication causes intense phytoplankton blooms in the coastal ocean. Sedimentation of the phytoplankton carbons will lead to their accumulation in the surficial sediment (Cloern, 2001), including PUFA compounds derived from the lipid production. Resuspension and oxidation of these PUFA-rich organic particles during ..."

**57. Literature cited**

Cloern, J. E. 2001. Our evolving conceptual model of the coastal eutrophication problem. Mar. Ecol.-Prog. Ser. 210: 223-253, doi.

Robinson, C., and P. J. Le B Williams. 2005. Respiration and its measurement in surface marine waters, p. 147-180. In P. A. del Giorgio and P. J. Williams, le B [eds.], Respiration in aquatic ecosystems. Oxford University Press.

**Response: Agree. The mentioned references have been cited in the revised manuscript.**

**Response to Anonymous Referee #2**

1. However, there appeared to be some problems about the experimental design of this study and the manuscript fails to provide convincing results.

Response: We have carefully addressed the reviewer's comments on our experimental design and the related data and results. Please refer to our detailed responses to each of these specific comments below.

2. More details about the motivation and experiment procedure should be included and clarified and the conclusions should be carefully justified.

Response: We thank the reviewer for this suggestion. We have carefully rewritten these parts of the manuscript in the introduction and the method sections to clarify the motivation of our study and provide the details of the experiment setup, operational procedure, and relevant methodology. We have also rewritten the conclusion section as the review suggested. Please refer to the specific comments of this reviewer below for details of our revisions in each section.

3. My major concerns about this manuscript is that the authors consider PAB as bacteria attached to particles with a size >25  $\mu$ m in the microcosm incubation. A great variety of bacteria would be lost, which would affect the major conclusion of the manuscript. The abundance of bacteria attached to >25  $\mu$ m particles would be significantly lower than that of free-living bacteria.

Response: We completely agree with the reviewer that a complete PAB community should be acquired using a smaller filtration such as 0.8  $\mu$ m. Actually, in the high turbid estuarine waters of the PRE, PAB on the particle size of > 25  $\mu$ m could be only about 20 percent of the PAB on the particle size of > 2  $\mu$ m (Ge et al., 2020). However, we should emphasize that our primary goal is to explore the mechanism for PUAs affecting PAB variation and associated oxygen consumption in high turbidity and low oxygen regions of the PRE. Although the concentration of PUAs was nanomolar in the bulk water, it can reach a micromolar level on the surface of the particles where they are produced. The hotspot PUAs concentration associated with particles is defined as the PUAs concentration in the volume of the water parcel displaced by these particles. As PUAs can be accumulated on larger particles (Edwards et al., 2015), we chose particle aggregates of > 25  $\mu$ m to perform the PUA-amended experiments, in order to better explore the PUAs effects on PAB in the hypoxic waters. Future study may need to investigate PUAs impacts on PAB associated with the particle size of 0.8-25  $\mu$ m.

4. Also, is this particle size proper for measurement of polyunsaturated aldehydes?

Response: It is specific for particle-adsorbed PUAs on particles of > 25  $\mu$ m. A previous study by Edwards et al (2015) uses an even larger size of 50  $\mu$ m for collecting sinking particles for PAB and

**the associated estimation of hotspot PUAs concentration.**

5. In addition, it is not clear why the authors choose 1 or 100  $\mu$ mol L-1 but not the background value for the incubation.

Response: We should emphasize that the concentration of PUAs in the water-column is inhomogeneous due to the presence of particles. Although the concentration of PUAs was nanomolar in the bulk water, it can reach a micromolar level on the surface of the particles where they are produced. The micromolar level of PUAs for incubation was chosen to represent the actual hotspot concentration of PUAs (the PUAs concentration in the volume of the water parcel displaced by the aggregation particles) not the mean PUAs concentration (nanomolar level) in the background seawater.

6. Moreover, bacterial community of the initial inoculates was lacking.

**Response: In the revised manuscript, we have provided the initial bacterial community data (T=0) for the experiment in the supplementary material (Figure S2).**

7. Personally, the most interesting part of this study is the role of polyunsaturated aldehydes-enhanced bacterial oxygen demand for the seasonal hypoxia. Thus, it is important to know to what extent different concentration of polyunsaturated aldehydes affect bacterial growth and respiration. Although the authors provide discussion on this, more analyses including the selection of background concentration of polyunsaturated aldehydes and testing on pure isolates are needed.

Response: We thank the reviewer for this suggestion. In the revised manuscript, we have provided a discussion on the impact of the background nanomolar level of PUAs on bacteria activity. "It should be mentioned that it remains controversial on the effect of background nanomolar PUAs on free-living bacteria, which is not our focus in this study. Previous studies suggested that 7.5 nmol  $L^{-1}$  PUAs would have a different effect on the metabolic activities of distinct bacterial groups in the NW Mediterranean Sea, although bulk bacterial abundance remained unchanged (Balestra et al., 2011). In particular, the metabolic activity of  $\gamma$ -Pro was least affected by nanomolar PUAs, although those of Bacteroidetes and Rhodobacteraceae were markedly depressed (Balestra et al., 2011). However, the daily addition of 1 nmol  $L^{-1}$  PUAs was found to not affect bacterial abundance and community composition during a mesocosm experiment in the Bothnian Sea (Paul et al., 2012)."

8. It is unclear why the authors studied the effect of polyunsaturated aldehydes of bacterial communities. The importance of polyunsaturated aldehydes was not properly and clearly presented. For examples, although the authors mentioned the effect of polyunsaturated aldehyde on marine microorganisms, detailed processes and mechanisms are not provided.

Response: We agree with the reviewer on this. In the revised manuscript, we have carefully rewritten this part by emphasizing the importance of PUAs on the microbial community and the associated mechanisms. "Phytoplankton-derived polyunsaturated aldehydes (PUAs) are known to affect marine microorganisms over various trophic levels by acting as infochemicals and/or by chemical defenses (Ribalet et al., 2008; Ianora and Miralto, 2010; Edwards et al., 2015; Franzè et al., 2018). PUAs are produced by stressed phytoplankters during the oxidation of membrane polyunsaturated fatty acids (PUFA) by lipoxygenase (Pohnert 2000) and are released from the surface of particles to the seawater by diffusion. The level of PUAs in the water-column are inhomogeneous, varying from sub-nanomolar offshore to nanomolar nearshore (Vidoudez et al., 2011; Wu and Li, 2016; Bartual et al., 2018), and to micromolar associated with particle hotspots (Edwards et al., 2015). The strong effect of PUAs on bacterial growth, production, and respiration has been well demonstrated in laboratory studies (Ribalet et al., 2008) and field studies (Balestra et al., 2011; Edwards et al., 2015). A perennial bloom of PUA-producing diatoms in the PRE mouth (Wu and Li, 2016) may indicate the relative importance of PUAs for microbial activity here compared to many other organic compounds, such as 2-n-pentyl-4-quinolinol (Long et al., 2003) and acylated homoserine lactones (Hmelo et al., 2011). A nanomolar level of PUAs recently reported in the coastal waters outside the PRE was hypothesized to affect oxygen depletion by promoting microbial utilization of organic matters in the bottom waters (Wu and Li, 2016). Meanwhile, the actual role of PUAs on bacterial metabolism within the bottom hypoxia remains largely unexplored."

9. Line 43: Please provide reference for the higher abundance of PAB compared to FLB.

Response: Done. The revised sentence is written as "In some coastal waters, PAB could be more abundant than the FLB with a higher metabolic activity to affect the coastal carbon cycle through organic matter remineralization (Garneau et al., 2009; Lee et al., 2015)."

10. Line 59-60: How polyunsaturated aldehyde affects marine microorganisms? Please provide more details.

Response: Done. We have rewritten the sentence as "Phytoplankton-derived polyunsaturated aldehydes (PUAs) are known to affect marine microorganisms over various trophic levels by acting as infochemicals and/or by chemical defenses (Ribalet et al., 2008; Ianora and Miralto, 2010; Edwards et al., 2015; Franzè et al., 2018). ".

11. Line 63: What is the meaning of affect oxygen depletion? Is this a promoting or inhibiting process?

**Response:** Done. The sentence has been rewritten as "... affect oxygen depletion by promoting microbial utilization of organic matters ...".

12. Line 72-73: It is strange to place this sentence here. Why PUAs did not serve as carbon source?

Response: Agree. We have moved the sentence to the method and result sections. Firstly, PUAs can be toxic to some bacteria precluding its use as a carbon source. Secondly, the specific arrangement of two double bonds and carbonyl chain make PUAs not a group of labile organic carbon for bacterial utilization. There were other studies supporting that PUAs could not serve as a carbon source for bacterial growth (Ribalet., 2008; Edwards et al., 2015).

13. Line 81: Is that "July  $2^{nd}$ "?

Response: It is July 2nd. We have corrected this typo in the revised manuscript.

14. Line 88: what do you mean by "pPUAs and dPUAs"?

**Response: Done. The abbreviations of pPUAs and dPUAs have been defined in the revised manuscript.**

15. Line 265: I did not see description of methods about the bacterial community analysis in bottom waters of X1, X2, X3 and PAB on particles of >25  $\mu$ m.

Response: Done. We have added the method descriptions of these data to the section of 2.7.4 in the revised manuscript. "DNA samples for the bulk bacteria (>0.2  $\mu$ m) and PAB on particles of >25  $\mu$ m at station Y1 were also collected for bacterial community analysis using the same method described above. Methods for the bulk water bacterial community analyses at station X1, X2, and X3 during the 2016 cruise can be found in the published paper of Xu et al. (2018)."

16. Line 276: Please provide the concentration of pPUAs and dPUAs.

**Response: Done. The mean concentrations of pPUAs and dPUAs have been provided in the revised manuscript.**

17. Line 278: shown Line 391: According to the results, low dose (1  $\mu$ mol) of PUAs can stimulate the growth of PAB, significantly different from that of high dose (100  $\mu$ mol) treatment. However, the test of PUAs as organic carbon source was conducted with 200  $\mu$ mol of PUAs. I guess such a high concentration would adversely affect bacteria growth, while the low dose PUAs is likely to be used as organic sources.

Response: The 200 µM PUAs used in the test of carbon source possibility was to assure the same level of organic carbon substrate as those for ALK and PAHs. Bacteria may need a longer time and a higher substrate concentration to utilize these refractory organic matters (ALK and PAHs).

**Although we have no test for the low-dose PUAs, the previous study has suggested that low-level PUAs (1 $\mu$ M and 10 $\mu$ M) were not used as a carbon source by bacteria (Edwards et al., 2015).**

18. Line 686: data of panel D are reproduced from Ribalet et al., 2008. Is this panel E? No methods were provided for growth of *Alteromonas hispanica* MOLA151.

**Response: Agree. It should be panel E. We have corrected this in the revised manuscript. We have also provided the growth method of A. hispanica MOLA151.**

19. Line 448: Since bacteria on >25  $\mu$ m particles can be low, hypothesis on signaling molecules may be tuned down.

[revised manuscript text omitted]
 10-uL mixture containing 1 uL 287 Toptag Buffer, 0.8  $\mu$ L dNTPs, 10  $\mu$ M primers, 0.2  $\mu$ L Tag DNA polymerase, and 1  $\mu$ L Template DNA. 288 289 Three parallel amplification products for each sample were purified by an equal volume of AMpure XP magnetic beads. Sample libraries were pooled in equimolar and paired-end sequenced ( $2 \times 250$  bp) on an 290 291 Illumina MiSeq platform.

High-quality sequencing data was obtained by filtering on the original off-line data. Briefly, the raw 292 data was pre-processed using TrimGalore to remove reads with qualities of less than 20 and FLASH2 to 293 294 merge paired-end reads. Besides, the data were also processed using Usearch to remove reads with a total base error rate of greater than 2 and short reads with a length of less than 100 bp and using Mothur to 295 296 remove reads containing more than 6 bp of N bases. We further used UPARSE to remove the singleton 297 sequence to reduce the redundant calculation during the data processing. Sequences with similarity greater than 97% were clustered into the same operational taxonomic units (OTUs). R software was used for 298 community composition analysis. 299

300 DNA samples for the bulk bacteria (>0.2 μm) and PAB on particles of > 25 μm at station Y1 were also
 301 collected for bacterial community analysis using the same method described above. Methods for the bulk
 302 water bacterial community analyses at stations X1, X2, and X3 during the 2016 cruise can be found in the
 303 published paper of Xu et al. (2018).

304

**305 2.8 Statistical Analysis**

306 All statistical analyses were performed using the statistical software SPSS (Version 13.0, SPSS Inc.,

307 Chicago, IL, USA). A student's t-test with a 2-tailed hypothesis was used when comparing PUAs-amended

308 treatments with the control or comparing stations inside and outside the hypoxic zone, with the null

309 hypothesis being rejected if the probability (p) is less than 0.05. We consider p of <0.05 as significant and p

of <0.01 as strong significant. Ocean Data View with the extrapolation model "DIVA Gridding" method

311 was used to contour the spatial distributions of physical and biogeochemical parameters.

312

313 **3. Results**

**314 **3.1** Characteristics of hydrography, biogeochemistry, and bulk bacteria community in the hypoxic**

315 **zone**

During our study periods, there was a large body of low oxygen bottom water with the strongest hypoxia (< 316 62.5  $\mu$ mol kg-1) on the western shelf of the PRE (Figure 1), which was relatively similar constant among 317 different summers of 2016 and 2019 (Figure 1). For vertical distribution, a strong salt-wedge structure was 318 319 found over the inner shelf (Figures 3A, 3D) with freshwater on the shore side due to intense river discharge. Bottom waters with oxygen deficiency (< 93.5 µmol kg-1) occurred below the lower boundary of the 320 salt-wedge and expanded ~60 km offshore (Figure 3E). In contrast, a surface high Chl-a patch (6.3  $\mu$ g L-1) 321 322 showed up near the upper boundary of the front, where there was enhanced water-column stability, low turbidity, and high nutrients (Figures 3B, 3C). Therefore, there was a spatial mismatch between the 323 subsurface hypoxic zone (Figure 3E) and the surface chlorophyll-bloom (Figure 3F) during the 324 estuary-to-shelf transect, as both the surface Chl-a and oxygen right above the hypoxic zones at the bottom 325 boundary of the salt-wedge were not themselves maxima. 326

There were much higher rates of respiration (BR) (t=7.8, n=9, p<0.01) and production (BP) (t=13.0, n=9, p<0.01) for the bulk bacterial community (including FLB and PAB) in the bottom waters of X1 within the hypoxic core than those of X2 and X3 outside the hypoxic zone during June 2016 (Figure 4, modified from data of Xu et al., 2018). The size-fractionated respiration rates were quantified at station Y1 during the 2019 cruise (Figure S1) to distinguish the different roles of FLB and PAB on bacterial respiration in the hypoxic waters. Our results suggested that bacterial respiration within the hypoxic waters was largely. contributed by PAB (>0.8 µm), which was about 2.3-3 folds of that by FLB (0.2-0.8 µm).

[revised manuscript text omitted]

---

## Author Response (AR2)

December 11, 2020

Dear Editor,

Thank you for the opportunity to revise the manuscript entitled "Impacts of biogenic polyunsaturated aldehydes on metabolism and community composition of particle-attached bacteria in coastal hypoxia" again.

We thank the editor and the reviewer for their constructive comments. We have carefully addressed them in revising the manuscript. Please see below our responses and the revised manuscript with tracked change. We hope that you will now find our manuscript suitable for publication.

Sincerely yours,

Qian Li
South China Sea Institute of Oceanology
Chinese Academy of Sciences, Guangzhou, China
Email: qianli@scsio.ac.cn

**Response to Editor**

Overall I think the manuscript is close to being ready for publication, but I would like to see you address the additional comments from Reviewer 1. Specifically, I would like to see you address the quantitative importance of PUAs in marine environments and the concentration of methanol in your treatments. Please make it clear where revisions to the manuscript were made when you resubmit.

**Response: 1) Quantitative importance of PUAs**
**To the best of our knowledge, there is no report in the literature that directly compares the levels of PUAs with other organic compounds in the same natural sample. It has only been known that PUA accounts for a small fraction (1-16%) of POC in sinking particles (Edward et al., 2015). Therefore, we follow the reviewer's suggestion by adding a statement to the relevant part of the introduction section (Page 4, Lines 66-68) as "…** **Phytoplankton-derived polyunsaturated aldehydes (PUAs) are known to affect marine microorganisms over various trophic levels by acting as infochemicals and/or by chemical defenses, which strengthen their potential importance in natural environments (Ribalet et al., 2008; Ianora and Miralto, 2010; Edwards et al., 2015; Franzè et al., 2018)** **…".**
**2) Methanol concentration in the treatments**
**The 0.05 % (v/v) of methanol is about 10 mmol $dm^{-3}$, which is higher than its natural level in seawater. But this concentration is lower than those used by several previous studies (e.g. Patterson and Ricke, 2015; Franzè et al., 2018). Consistent with these studies, we did not see a much difference of bacterial community by the methanol level of 0.05%. We have added a comment on this to the method section of the revised manuscript (Page 10, Lines 208-210) as "…** **We should note that the methanol percentage (0.05% v/v) here is higher than its natural level in seawater although no substantial change of bacteria community was found…"**

**Response to Reviewer**

1. Regarding the importance of PUA it is clear to me from the replies and revision that it is the signaling properties of PUA (rather than as a substrate) that is referred to regarding its ecological importance (added in the introduction). This is a valid statement. Still, I cannot see a reference or argument showing the PUAs are quantitatively important in relation to other organic compounds (in either way) by challenging them in the same sample under natural conditions (cf. my comment nr. 2). A solution is to just state that PUAs are of potential importance in the natural environment (as shown by the references), without direct comparison with other compounds.

**Response: To the best of our knowledge, there is no report in the literature that directly compares the levels of PUAs with other organic compounds in the same natural sample. It has only been known that PUA accounts for a small fraction (1-16%) of POC in sinking particles (Edward et al., 2015). Therefore, we follow the reviewer's suggestion by adding a statement to the relevant part of the introduction section (Page 4, Lines 66-68) as "… Phytoplankton-derived polyunsaturated aldehydes (PUAs) are known to affect marine microorganisms over various trophic levels by acting as infochemicals and/or by chemical defenses, which strengthen their potential importance in natural environments (Ribalet et al., 2008; Ianora and Miralto, 2010; Edwards et al., 2015; Franzè et al., 2018) …"**

2. Regarding the cut off size for particles, it may just be stated and motivated why 25 µm pore size was used, while also admitting that this may not be comparable to other cut offs used in the literature. The research field should benefit from finding a common practice.

**Response: Agree. We have rewritten the relevant sentence in the introduction section (Page 5, Lines 99-101) as "… We focused on particles of >25 µm to explore the role of PUAs on PAB associated with sinking aggregates and large suspended particles (it may not be directly comparable to other size cut-offs in the literature) …"**

3. My comment 28 in the previous review: Please make a calculation how much 0.05 % of methanol mean in final conc. of nmol dm-3? In my experience %-levels of solvents can mean high concentration as compared to natural levels of relevant substrates or toxic levels. If it is of potential influence I suggest to add a comment regarding that.

**Response: The 0.05 % (v/v) of methanol is about 10 mmol dm$^{-3}$, which is higher than**

its natural level in seawater. But this concentration is lower than those used by several previous studies (e.g. Patterson and Ricke, 2015; Franzè et al., 2018). Consistent with these studies, we did not see a much difference of bacterial community by the methanol level of 0.05%. We have added a comment on this to the method section of the revised manuscript (Page 10, Lines 208-210) as "… 
[revised manuscript text omitted]

[Figure]

**Figure 1**

[Figure]

**Figure 2**

[Figure]

**Figure 3**

[Figure]

**Figure 4**

[Figure]

**Figure 5**

[Figure]

**Figure 6**

[Figure]

**Figure 7**

[Figure]

**Figure 8**

[Figure]

M+PAH          M+ALK          M+C7_PUA

**Figure 9**